# LLM Discussion: Enhancing the Creativity of Large Language Models via Discussion Framework and Role-Play

**Li-Chun Lu,** * **Shou-Jen Chen,** * **Tsung-Min Pai, Chan-Hung Yu, Hung-yi Lee, Shao-Hua Sun**
Department of Electrical Engineering, National Taiwan University
{b08901207,b09901116,b09602017,r12942147,hungyilee,shaohuas}@ntu.edu.tw

## Abstract

Large language models (LLMs) have shown exceptional proficiency in natural language processing but often fall short of generating creative and original responses to open-ended questions. To enhance LLM creativity, our key insight is to emulate the human process of inducing collective creativity through engaging discussions with participants from diverse backgrounds and perspectives. To this end, we propose *LLM Discussion*, a three-phase discussion framework that facilitates vigorous and diverging idea exchanges and ensures convergence to creative answers. Moreover, we adopt a role-playing technique by assigning distinct roles to LLMs to combat the homogeneity of LLMs. We evaluate the efficacy of the proposed framework with the Alternative Uses Test, Similarities Test, Instances Test, and Scientific Creativity Test through both LLM evaluation and human study. The results show that our proposed framework outperforms single-LLM approaches and existing multi-LLM frameworks across various creativity metrics. The code is available at https://github.com/lawraa/LLM-Discussion.

## 1 Introduction

Large language models (LLMs) have emerged as highly efficient tools in addressing daily challenges across various applications, demonstrating exceptional capabilities in natural language processing (Lappuin, 2024; Yang et al., 2023b). Specifically, LLMs achieve excellent performance in language comprehension tasks, such as sentiment analysis (Zhang et al., 2023a) and question answering (Qin et al., 2023). Also, LLMs are widely adopted in content generation tasks, ranging from writing articles to composing poetry, by producing coherent and contextually relevant text that closely mimics human-written content (Qin et al., 2023; Yuan et al., 2022). Nevertheless, recent studies suggest that LLMs demonstrate limited creativity in answering open-ended questions and often fail to produce original responses (Ippolito et al., 2022; Chakrabarty et al., 2023; Mohammadi, 2024).

Psychological studies suggest that engaging in discussions with participants from diverse backgrounds, perspectives, and experiences can significantly enrich individual creativity (Han et al., 2022; Paulus & Nijstad, 2003; McGrath, 1984; Sutton & Hargadon, 1996; Karwowski & Soszynski, 2008). Can LLMs likewise strengthen their creative capabilities through collaborative discussions with diversified peers? While Du et al. (2023); Liu et al. (2023b); Sun et al. (2023) explore collaboration among LLMs, most existing works are limited to improving the performance on close-ended tasks, such as Massive Multitask Language Understanding (Hendrycks et al., 2021), Mathematics (Cobbe et al., 2021), and Code Generation (Huang et al., 2023), leaving discussions on creativity largely under-investigated. On the other hand, prior works researching LLM creativity are mostly restricted to a single LLM agent (Gómez-Rodríguez & Williams, 2023; Stevenson et al., 2022).

Our goal is to boost the creativity in LLMs by designing a discussion framework. Subsequently, two main challenges arise due to the nature of LLMs. First, most modern

---

*Equal contribution
Correspondence to: Shao-Hua Sun <shaohuas@ntu.edu.tw>

commercialized LLMs are not specifically trained or reinforced to engage in multi-turn conversations (Ouyang et al., 2022; Ding et al., 2023), which results in unengaging discussions among LLMs that quickly converge. Second, the high homogeneity of LLMs renders it difficult for discussions to diverge and produce creative outcomes (Ouyang et al., 2022; Padmakumar & He, 2024).

To address these issues, we develop a role-play enhanced LLM discussion framework, dubbed *LLM Discussion*, as illustrated in Figure 1. To facilitate vigorous and diverging exchanges throughout rounds of discussion and ensure convergence to creative answers, we arrange a three-phase discussion and design prompts specialized in the initiation phase, the discussion phase, and the convergence phase. Then, to tackle the LLM homogeneity, we adopt the role-playing technique, an established practice in Design Thinking (Camacho, 2016). Specifically, we propose to assign a diverse set of roles with distinct backgrounds and perspectives to LLM agents participating in a discussion, such as *Visionary Millionaire* and *Futurist*, and roles specialized for various facets of discussion from Six Thinking Hats methodology (Setyaningtyas & Radia, 2019).

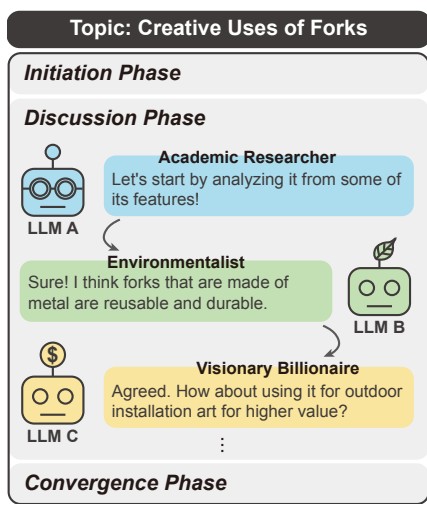

Figure 1: **Role-Play Enhanced LLM Discussion.**

To evaluate our proposed framework, we incorporate four creativity benchmarks, covering a broad spectrum of creative thought processes: Alternative Uses Test, Instances Test, and Similarities Test from the Wallach-Kogan Creativity Test (WKCT) (Wallach, M. A., and Kogan, N., 1965), and Scientific Creativity Test (Hu & Adey, 2002). We conduct LLM evaluation and human study on four widely used metrics: *Originality*, *Elaboration*, *Fluency*, and *Flexibility* (Torrance, 1966). The results demonstrate that our proposed framework outperforms baselines in producing creative answers.

## 2 Related Work

**Multi-LLM Collaboration.** Prior works have developed multi-LLM frameworks to improve the factuality, reasoning, arithmetic skills, and coding abilities of LLMs (Du et al., 2023; Liang et al., 2023; Chan et al., 2024; Wu et al., 2023; Li et al., 2023c; Liu et al., 2023b; Ning et al., 2024; Jiang et al., 2023; Sun et al., 2023), mostly evaluated on MMLU (Hendrycks et al., 2021), GSM8K (Cobbe et al., 2021), and HumanEval (Chen et al., 2021). In contrast, our work aims to explore enhancing LLM creativity by designing a multi-LLM discussion framework.

**LLM Creativity.** Recently, many aspects of LLM creativity (Chakrabarty et al., 2023; Ippolito et al., 2022) have been studied, such as English creative writing (Gómez-Rodríguez & Williams, 2023), assessing metaphors (Distefano et al., 2024), and coming up with alternative uses of objects (Stevenson et al., 2022). These studies are mainly limited to single-LLM setups; instead, this work aims to employ multiple LLMs to induce collective creativity.

**LLM Role-Playing.** LLMs have shown significant potential for role-playing (Wang et al., 2023c; Park et al., 2023; Li et al., 2023a; Wei et al., 2023; Shanahan et al., 2023; Salemi et al., 2023). The ability to impersonate distinct roles has proven to induce human-like behavior (Park et al., 2023; Wang et al., 2023b; Li et al., 2023a; Shao et al., 2023), interactivity (Wang et al., 2023c; Cui et al., 2023), and the ability to tackle complex tasks of LLMs (Cui et al., 2023; Li et al., 2023a; Wang et al., 2023c). Our work aims to harness LLM's ability to role-play to enhance LLM creativity.

# 3  Approach: LLM Discussion

Our key insight is to boost creativity in LLMs by designing an effective discussion framework and employing role-play techniques. To this end, we propose *LLM Discussion*, a three-phase discussion framework that incorporates role-playing, allowing multiple LLM agents to build on each other's thoughts and collectively produce creative outcomes.

Specifically, to combat the inability of LLMs to conduct engaging multi-turn discussions, we propose to arrange three different phases, the initiation, discussion, and convergence phases, as described in Section 3.1. In each phase, a short-term objective is provided to each LLM with a specialized prompt, *e.g.*, actively discuss or efficiently converge, facilitating both divergent and convergent thinking. Then, to alleviate the LLM homogeneity issue, *i.e.*, all LLMs think alike, Section 3.2 presents our proposed role-play mechanism, which assigns diverse backgrounds, perspectives, and personalities to LLMs.

## 3.1  Discussion Framework

We aim to design a discussion framework that hosts rounds of dynamic exchanges of ideas, progressively fostering novel solutions and deeper insights. However, most modern commercialized LLMs are not specifically trained or reinforced to engage in multi-turn conversations (Ouyang et al., 2022; Ding et al., 2023), resulting in unengaging discussions among LLMs that quickly converge. To facilitate meaningful interactions among LLMs, Du et al. (2023) introduce LLM Debate, which requires multiple LLMs to propose and debate their individual responses with other LLMs over multiple rounds to arrive at a joint answer agreed upon by all the LLMs.

Yet, LLM Debate is devised to improve LLM's performance in answering closed-ended questions, such as question answering (Hendrycks et al., 2021) and mathematical reasoning (Cobbe et al., 2021). In particular, the mechanism employed in LLM Debate requires each LLM to maintain its own answer while observing answers produced by other LLMs, which is effective in resolving factual questions but can fall short of promoting divergent thinking building upon others' responses.

To encourage LLMs to discuss with others and inspire others actively to engender collective creativity, we devise a three-phase discussion framework that explicitly requires each LLM to build upon others' responses. We arrange an initiation phase, a discussion phase, and a convergence phase, which are described in detail below. In each phase, each LLM aims to fulfill an objective, *e.g.*, actively discuss or efficiently converge, as instructed by a specialized prompt. The proposed discussion framework is illustrated in Figure 2.

**Initiation Phase.** This phase serves as the groundwork for the discussion, where each LLM is introduced to the topic and the objectives of the dialogue. In this phase, we aim to explicitly inform the LLMs about the situation and declare the goal. Specifically, we notify each LLM that it is in a group discussion setup by stating "you are in a group discussion with other teammates" in the prompt. Moreover, we stress the importance of teamwork spirit with the instruction "Initiate a discussion with others to collectively complete the following task." Then, we provide a description of the task that we want the LLMs to solve, *e.g.*, "come up with creative uses for forks."

**Discussion Phase.** The goal of this phase is to induce multiple rounds of discussions carrying meaningful idea exchanges among LLMs. In each discussion round, each LLM receives responses from all other agents in the previous round, resembling the idea of active listening in human discussions. Additionally, we aim to encourage each LLM to compose and diverge its answers. For each LLM to balance between generating its own answers and following up on others' thoughts, we experiment with various prompts as discussed in Section 4.5.

**Convergence Phase.** After rounds of diverging and creative discussions, the convergence phase aims to summarize the ideas brought in discussions and converge to a collective conclusion. To this end, we inform each LLM by prompting it with the instruction, "This is the last round of the discussion; please finalize and present a list of creative answers." Note

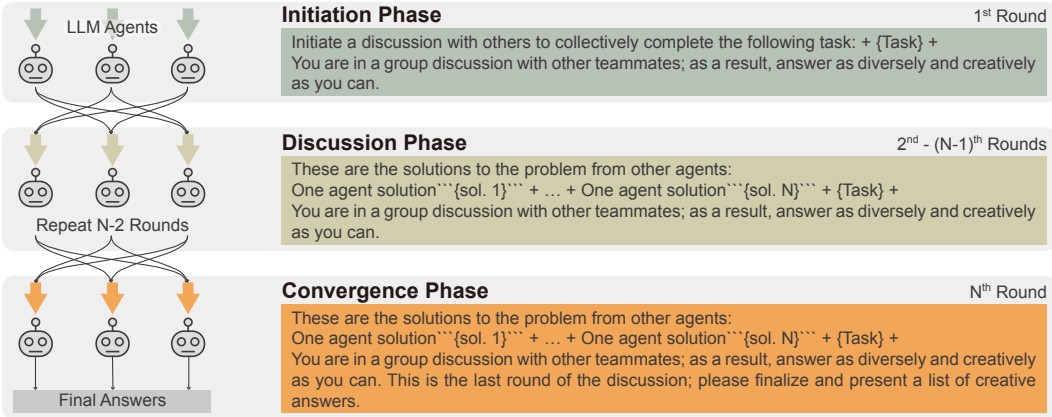

Figure 2: **Discussion Framework.** We propose an LLM discussion framework that induces collective creativity by bringing divergent and convergent thinking together. The initiation phase informs LLMs of the discussion setup and the task. The discussion phase allows LLMs to build on the ideas of others as well as diverge and generate their own answers. The convergence phase summarizes the discussed ideas and draws a collective conclusion.

that the term "finalize" in our convergence prompt serves as an explicit prompt to conclude, while "present a list" facilitates parsing.

**Full Discussion Procedure.** Putting everything together, our proposed discussion procedure with $N$ rounds is described as follows.

- $1^{st}$ Round: In the first discussion round, we initiate the discussion by prompting LLMs using the prompt described in the initiation phase.

- $2^{nd}$ to $(N-1)^{th}$ Rounds: To encourage multiple rounds of discussions, we repetitively prompt each LLM and provide it with responses from other LLMs using the prompt presented in the discussion phase.

- $N^{th}$ Round: The goal of the last discussion round is to summarize the ideas and converge to a collective conclusion. Hence, the LLMs are prompted with the convergence phase prompt. Note that we also provide the context by first prompting LLMs with the discussion phase prompt.

A five-round discussion qualitative result is shown in Section K.

## 3.2 Role-Play

We empirically observe that even with engaging discussions, LLMs often fall short of producing diverging answers given the same prompts due to their homogeneity, which aligns with recent findings reported by Ouyang et al. (2022); Padmakumar & He (2024). To address this issue, we propose to employ role-play, a technique commonly used in the ideation stage of Design Thinking (Camacho, 2016). This technique aids people in coming up with innovative ideas for various purposes, unlike problem-oriented roles, which focus on solving specific issues within distinct fields.

**Role Generation.** To generate a set of roles with diverse backgrounds, perspectives, and experiences and detailed descriptions of these roles, we propose an automated pipeline that uses GPT-4 (OpenAI et al., 2023) to produce a list of {**role**, **speciality**, **role prompt**}, *e.g.*, {*Visionary Millionaire*, *Financial success and forward-thinking*, *As a Visionary Millionaire, your mission is to leverage your financial insight and forward-thinking approach to inspire groundbreaking ideas ...*}. The detail process and generation prompts are presented in Section J. We also adopt Six Thinking Hats methodology (Setyaningtyas & Radia, 2019) to give LLMs different perspectives, *e.g.*, the red hat represents emotions, and the green hat stands for innovation.

The entire role set can be found in Section B. Our work makes an initial attempt to automatically generate diverse and detailed roles for discussion, and systematically exploring other role-generation mechanisms is left for future work.

**Role-Play Enhanced Discussion.** To incorporate the roles into our discussion framework, we assign different roles to different LLMs before the initiation phase. Specifically, we adopt an autoregressive way to select the roles for each member in a discussion. Then, at the beginning of each discussion round, we remind LLMs of their roles and responsibilities and ask them to declare their roles. This ensures that other members of the discussion are aware of each speaker's identity, enabling a more structured and coherent exchange of ideas. Note that the roles of LLMs stay unchanged throughout the entire discussion. An illustration of role assignment and examples of roles are shown in Figure 3.

**Speciality**: Financial Success and Forward-Thinking
**Role Prompt**: As a Visionary Millionaire, your mission is to leverage your financial insight and forward-thinking approach to inspire groundbreaking ideas. Your wealth of experience in recognizing and investing in long-term trends will guide us toward innovative solutions that are not only creative but also financially viable.

Figure 3: **Role Assignment and Example.** At the beginning of a discussion, each LLM is assigned a role with *specialties* and a detailed description of the role, *i.e.*, *role prompt*.

## 4 Experiment

### 4.1 Benchmarks

Creativity encompasses several distinct aspects, including divergent thinking, connection-making, and practical innovation. To measure LLM creativity, we utilize four existing renowned benchmarks for human creativity in our evaluation, each responsible for assessing distinct aspects of creativity. We adopt three benchmarks from the widely-used Wallach-Kogan Creativity Tests (Wallach, M. A., and Kogan, N., 1965): Alternative Uses Task (AUT), Instances Test (INSTANCES), and Similarities Test (SIMILARITIES). We also include the Scientific Creativity Test (Hu & Adey, 2002) (SCIENTIFIC), which measures scientific creativity and scientific process skills.

AUT assesses the ability to develop practical and innovative use cases for objects like a fork or a shoe. INSTANCES requires participants to list things following specific criteria *e.g.*, things that move on wheels or things that are square. SIMILARITIES evaluates creativity in making connections, challenging participants to brainstorm and link ideas through commonalities, *e.g.*, how a book and a magazine are alike. Lastly, SCIENTIFIC evaluates creativity in scientific contexts and the ability to devise practical and creative solutions to scientific problems. Some sample tasks of these benchmarks are presented in Table 1.

**Task augmentation.** Due to the limited public availability of the tasks from these benchmarks, we generate additional tasks using GPT-4 (OpenAI et al., 2023). Specifically, for each benchmark, we provide GPT-4 with publicly available task samples and ask it to generate 30 additional tasks (Liu et al., 2023a; Ubani et al., 2023; Fang et al., 2023; Peng et al., 2023). Then, we carefully review the generated tasks to ensure their relevance and alignment with the benchmarks. We aggregate the original and generated tasks to form each benchmark for evaluation. We will release the full task sets to facilitate future research studying LLM creativity.

### 4.2 Baseline

We compare our proposed framework, LLM Discussion, with the following methods.

- **Single Agent** involves querying a single LLM agent to solve the task on its own. We experiment with various prompting techniques, such as few-shot prompting (Brown et al., 2020), zero-shot-CoT (Kojima et al., 2022), LLM Stimuli (Li et al., 2023b), and the take-a-deep-breath prompt (Yang et al., 2023a), and the results are shown in Table 5. Since no particular prompt consistently outperforms others, we select the zero-shot prompt, *i.e.*, with only task description, as our main baseline.

| BENCHMARK | DESCRIPTION | SAMPLE TASK |
|---|---|---|
| AUT | This benchmark requires participants to come up with as many creative uses as possible for a simple object. | What are some creative uses for a fork? |
| INSTANCES | This benchmark requires participants to list as many creative answers as possible that fit a specific criterion. | Name all the round things you can think of. |
| SIMILARITIES | This benchmark asks participants to creatively explore similarities between two things. | Tell me all the ways in which a book and a magazine are alike. |
| SCIENTIFIC | This benchmark asks participants to generate innovative ideas, envision hypothetical scenarios, propose improvements, and design experiments, all scientifically. | Please think of as many possible improvements as you can to a regular bicycle, making it more interesting, more useful, and more beautiful. |

Table 1: **Benchmark Descriptions and Sample Tasks.** Alternative Uses Task (AUT), Instances Test (INSTANCES), and Similarities Test (SIMILARITIES) are designed to assess creative thinking in everyday contexts, while the Scientific Creativity Test (SCIENTIFIC) evaluates scientific creativity and scientific process skills.

- **Brainstorm, then Select (BTS)** is specifically designed for improved originality and utility on AUT proposed by Summers-Stay et al. (2023).

  It includes a selection phase where potential solutions are judged against specific criteria to identify the most original responses. Since it is designed for AUT, we only present the AUT results of this method. Also, BTS is a single-LLM framework, unlike our proposed framework, which encourages iterative enhancements through multi-round discussions and dynamically evolving the brainstorming process rather than filtering static responses.

- **LLM Debate** (Du et al., 2023) is a multi-LLM framework aiming to improve factuality and reasoning ability of LLMs by requiring LLMs to debate their responses and reasoning processes over multiple rounds. Specifically, the LLM Debate asks each LLM to examine responses from other LLMs and verify the correctness and reasonableness instead of following up on the ideas brought up by others. In contrast, our LLM Discussion framework focuses on developing collaborative discussions with LLMs assigned diverse roles, allowing for the emergence of collective creativity.

More details of these methods can be found in Section A.

### 4.3 Evaluation

We use LLMs and humans to quantitatively evaluate the creativity of generated responses.

#### 4.3.1 Metrics

We employ the metrics established by the Torrance Tests of Creative Thinking (TTCT) (Torrance, 1966), recognized for their robustness in measuring creative capabilities and are commonly applied in assessing human creativity (Cramond et al., 2005). The TTCT framework comprises four distinct metrics:

- *Originality:* Considering novelty responses, not familiar and unusual, but relevant.
- *Elaboration:* The amount of details used to extend a response.
- *Fluency:* The number of relevant responses.
- *Flexibility:* A variety of categories or shifts in responses.

**Originality** and **Elaboration** provide qualitative measures crucial for evaluating the depth and novelty of responses, which reflects the creative potential of LLMs in a 5-point Likert scale. On the other hand, **Fluency** and **Flexibility** are measured cumulatively, aggregating the amount of relevant answers or categories; yet, unlike humans, LLMs are inherently

capable of generating large volumes of responses (OpenAI et al., 2023), making **Fluency** and **Flexibility** less indicative of creativity for LLMs. Therefore, **Originality** and **Elaboration** are more informative and can more accurately reflect LLM creativity, while **Fluency** and **Flexibility** results are still presented to ensure adherence to the TTCT standards. Section G provides more detailed explanations.

### 4.3.2 LLM Evaluation

LLMs have demonstrated the ability to resemble human judgment and reasonably evaluate contents, achieving results comparable to human (Chan et al., 2024; Wang et al., 2023a; Adlakha et al., 2023; Kocmi & Federmann, 2023). As a result, we adopt LLMs to evaluate the creativity of generated responses. Specifically, we use ChatGPT (Wang et al., 2023a) for the LLM evaluation.

In our LLM evaluation, responses are assessed individually on Originality and Elaboration and collectively for Fluency and Flexibility. That said, the response to "*Creative uses of a fork*" might be [*Eating Pasta*, *Eating Pancake*, *Art projects*, *Drilling*]. For Originality and Elaboration, each response receives an individual score, for example, [2, 2, 3, 1] and [2, 2, 2, 1], respectively. For Fluency and Flexibility, the entire list of responses is collectively evaluated three times, receiving an average score, for instance, 3.33 and 2.33, respectively.

### 4.3.3 Human Evaluation

We also conduct human evaluation on the responses generated by our method and the baselines. Human evaluation is conducted using the same metrics as those used in LLM evaluation. To ensure consistency across both setups, the same rubric used in the LLM evaluations is also presented to human evaluators, as shown in Section C. In total, 1,406 responses from 42 distinct annotators are collected online for evaluating the creativity of generated responses. Additionally, we investigate the correlation between human and LLM evaluations in Section 4.4.2.

## 4.4 Results

Our experiments and evaluation both utilize the gpt-3.5-turbo-0125 model.

### 4.4.1 LLM Evaluation Results

We present the LLM evaluation results across four benchmarks in Table 2. Our proposed framework, LLM Discussion, outperforms the baselines (Single Agent, Brainstrom, then select, and LLM Debate) in Originality on four benchmarks and Elaboration on three benchmarks. In the following, we discuss the qualitative results observed in the chat logs provided in Section K from three perspectives.

**Collaborative Dynamics.** Our three-phase discussion framework stimulates a more collaborative tone; an example sentence is *"Building on those ideas, I believe we can combine..."* This results in more distinctive responses generated in each round, reducing the LLM homogeneity issue. In contrast, LLM Debate tends to be more discriminative, merely collecting or correcting previous responses without following up on ideas brought up by others.

**Role-Specific Responses.** Our proposed role-play technique encourages responses specifically aligned with the assigned roles, diversifying the creativity from the perspectives of various fields rather than being limited to general purposes. For example, in an AUT scenario of an *umbrella*, the role, *Futurist*, proposes integrating it with VR technology, while the role, *Environmentalist*, suggests using it as a shelter for wildlife. In contrast, Single Agent and LLM Debate tend to suggest more general uses, such as re-purposing the umbrella as a tray or as a quirky headpiece for a costume.

**Conceptual Complexity and Attention to Detail.** Our proposed LLM Discussion achieves superior performances in Elaboration by demonstrating expertise in developing more complex concepts and extending them with details. For example, when asked *"Name all the things you can think of that are used in culture"* from INSTANCES, Single Agent and LLM

| BENCHMARK | METHOD | ORIGINALITY | | ELABORATION | | FLUENCY | | FLEXIBILITY | |
|---|---|---|---|---|---|---|---|---|---|
| | | Mean | Std. | Mean | Std. | Mean | Std. | Mean | Std. |
| AUT | Single Agent | 3.47 | 0.38 | 3.08 | 0.39 | 8.99 | 1.10 | 8.82 | 1.49 |
| | Brainstorm, then Select | 3.84 | 0.61 | 3.32 | 0.65 | 4.63 | 1.43 | 4.60 | 1.63 |
| | LLM Debate | 3.73 | 0.47 | 3.78 | 0.47 | 10.47 | 2.96 | 9.63 | 2.73 |
| | LLM Discussion | **4.44** | 0.30 | **4.22** | 0.27 | 9.19 | 2.25 | 9.68 | 1.92 |
| INSTANCES | Single Agent | 2.46 | 0.33 | 1.89 | 0.29 | 14.32 | 4.52 | 5.82 | 3.11 |
| | LLM Debate | 2.61 | 0.32 | 1.90 | 0.29 | 26.21 | 6.73 | 11.28 | 8.72 |
| | LLM Discussion | **3.65** | 0.34 | **2.20** | 0.58 | 16.88 | 10.04 | 11.11 | 5.26 |
| SIMILARITIES | Single Agent | 2.66 | 0.39 | 1.99 | 0.31 | 7.00 | 1.76 | 6.49 | 1.61 |
| | LLM Debate | 2.81 | 0.21 | **2.61** | 0.35 | 9.65 | 2.34 | 8.32 | 2.24 |
| | LLM Discussion | **3.29** | 0.30 | 2.52 | 0.54 | 7.27 | 2.13 | 8.14 | 2.04 |
| SCIENTIFIC | Single Agent | 3.18 | 0.38 | 2.77 | 0.51 | 6.37 | 2.35 | 6.06 | 2.31 |
| | LLM Debate | 3.52 | 0.38 | 3.45 | 0.47 | 6.91 | 3.35 | 6.75 | 2.28 |
| | LLM Discussion | **3.95** | 0.25 | **3.47** | 0.55 | 5.58 | 2.61 | 5.91 | 2.39 |

Table 2: **LLM Evaluation Results.** The Originality and Elaboration scores from the LLM Discussion nearly surpass all the baselines with more than one standard deviation across four benchmarks: AUT, INSTANCES, SIMILARITIES, and SCIENTIFIC. The highest Originality and Elaboration scores in each benchmark are highlighted in bold.

Debate generate responses such as *Art*, *Music*, and *Clothing* with no further elaboration. Conversely, LLM Discussion generates concrete yet detailed concepts such as *Tattoos*, *Digital art*, and *Ethical fashion*, each accompanied by a detailed explanation.

The qualitative results from the AUT and SCIENTIFIC benchmarks presented in Figure 4 show that our LLM Discussion received higher Originality scores by producing more creative responses compared to other frameworks. The impact of temperature on creativity score is further discussed in Section H.

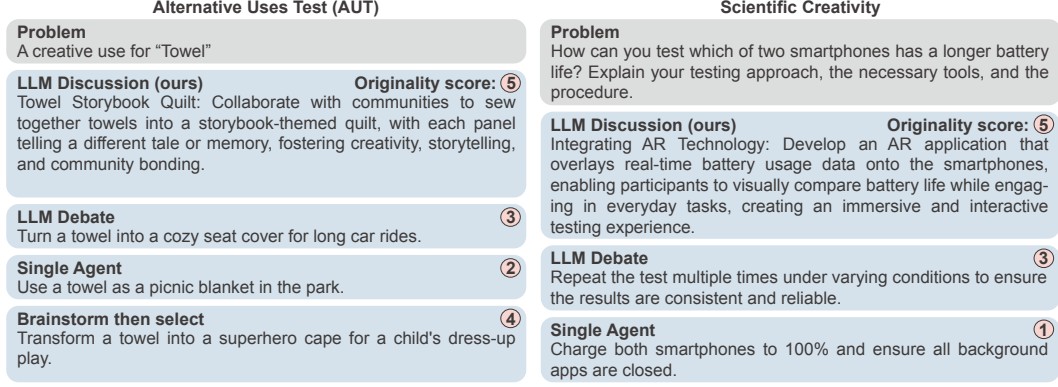

Figure 4: **Qualitative Results.** We present responses generated by LLM Discussion and other baselines on two benchmarks, AUT and SCIENTIFIC, along with the Originality scores from LLM evaluation. It demonstrates that ideas generated through LLM Discussion are more innovative and provide greater detail.

### 4.4.2 Human Evaluation and Correlation

Human evaluation results presented in Table 3 show that our proposed LLM discussion outperforms the baselines in Originality and Elaboration across three out of four benchmarks, aligning with the LLM evaluations.

We investigate the correlation between the LLM evaluation and human evaluation by calculating Kendall's $\tau$ correlation coefficient between LLM and human evaluations, and each human evaluator against the average ratings of the others human evaluators, as shown in Table 4.

| BENCHMARK | METHOD | ORIGINALITY | | ELABORATION | | FLUENCY | | FLEXIBILITY | |
|---|---|---|---|---|---|---|---|---|---|
| | | Mean | Std. | Mean | Std. | Mean | Std. | Mean | Std. |
| AUT | Single Agent | 2.50 | 1.24 | 1.93 | 0.98 | 8.13 | 2.01 | 4.33 | 1.95 |
| | LLM Debate | 3.80 | 1.17 | 2.50 | 0.96 | 7.25 | 2.99 | 4.38 | 2.45 |
| | LLM Discussion | **3.84** | 1.01 | **3.18** | 0.94 | 7.88 | 1.83 | 5.19 | 2.24 |
| INSTANCES | Single Agent | 1.67 | 0.85 | 1.48 | 0.91 | 8.55 | 1.99 | 4.50 | 2.36 |
| | LLM Debate | 1.25 | 0.66 | 1.50 | 0.71 | 7.20 | 1.60 | 4.90 | 1.45 |
| | LLM Discussion | **3.46** | 1.00 | **3.18** | 1.23 | 7.25 | 2.57 | 5.20 | 2.66 |
| SIMILARITIES | Single Agent | 1.97 | 1.05 | 2.10 | 0.90 | 4.42 | 0.64 | 3.33 | 0.85 |
| | LLM Debate | 1.90 | 1.30 | 2.38 | 0.48 | 8.50 | 0.76 | 4.50 | 2.06 |
| | LLM Discussion | **2.96** | 1.02 | **3.54** | 1.13 | 5.25 | 2.09 | 3.42 | 2.10 |
| SCIENTIFIC | Single Agent | 2.20 | 1.03 | 2.88 | 1.05 | 5.04 | 2.53 | 3.86 | 2.13 |
| | LLM Debate | **3.00** | 1.15 | **4.00** | 1.00 | 5.00 | 1.10 | 4.40 | 2.01 |
| | LLM Discussion | 2.95 | 1.41 | 3.70 | 1.10 | 3.57 | 0.73 | 3.54 | 1.48 |

Table 3: **Human Evaluation Results.** Our proposed framework, LLM Discussion, outperforms the baselines in Originality and Elaboration on three out of four benchmarks.

| KENDALL'S $\tau$ | ORIGINALITY | ELABORATION | FLUENCY | FLEXIBILITY |
|---|---|---|---|---|
| LLM - Human Average | 0.5213 | 0.2753 | 0.6017 | 0.5508 |
| Human - Human Average | 0.5094 | 0.4753 | 0.3692 | 0.2071 |

Table 4: **Correlation of Evaluation Results.** Kendall's $\tau$ correlation coefficient reveals a strong correlation between LLM evaluations and the average human evaluations in Originality, Fluency, and Flexibility, with a moderate correlation observed in Elaboration. Additionally, we present the correlation between individual human evaluations and the average of others, which exhibits high variance.

Note that $|\tau| \in [0.3, 1.0], [0.2, 0.3), [0.1, 0.2),$ and $[0, 0.1)$, are considered a strong, moderate, weak, and very weak correlation, respectively (Botsch, 2011; Chiang & yi Lee, 2023; Zhang et al., 2023b). The results show a strong correlation between LLM and the average human evaluation, except for Elaboration, where only a moderate correlation is observed.

In particular, for the SIMILARITIES, the Elaboration scores from human evaluations exceed those from LLM evaluations. Consider the response to the prompt, *"Tell me one way in which a donkey and a horse are alike"*, where the answer provided was: *"Interstellar Companions: Looking towards interstellar exploration, these animals might pioneer new frontiers in space travel, accompanying humans on missions and serving as invaluable companions and assistants in uncharted realms of the cosmos."* This response was scored 1 point on Elaboration by LLM evaluators but received an average of 3.88 from ten human evaluators. The potential reason is that human evaluators tend to favor lengthier responses, as shown in Section E, whereas LLM evaluations focuses on whether the response addresses the question and prioritize the development of ideas over mere verbosity. We also analyze the average length of the answers and their impacts on each benchmarks for Elaboration and Originality in Section F.

Interestingly, the correlation coefficients of LLM to the average human scores were higher than those among human evaluators, suggesting that human scores have a higher variance. This indicates that LLM evaluations are more correlated with the average human score than individual human evaluations are with the average human score.

### 4.5 Ablation Study

We conduct ablation studies in AUT on different prompts, the number of discussion rounds, and the number of LLM agents involved in discussions to determine the settings for the main experiments. Also, we verify the effectiveness of role-play and 3-phase discussion framework respectively.

**Role-Play and 3-Phase Discussion Framework.** We study the effect of role-play and discussion prompts individually. Section I shows the results of three different conditions of multi-LLM frameworks on each task: role-play without 3-phase discussion prompts,

3-phase discussion prompts without role-play, and both 3-phase discussion and role-play prompts (LLM Discussion).

Solely role-play or 3-phase discussion received better scores in Originality and Elaboration compared to single agent and LLM Debate. LLM Discussion, which features both discussion framework and role-play, has the best performance among the three, outperforming the other two across four out of eight Originality or Elaboration of each benchmark.

**Prompt Design.** We design various specialized prompts for our proposed three-phase discussion framework, aiming to explore how the prompts affect creativity in a discussion. We evaluate the prompt designs and present the results in Section D, which determines the prompt as the main prompt for further experiments.

**Rounds of Discussion.** We examine the effect of the number of discussion rounds and present the results in Figure 5. We can observe that Originality increases but Elaboration decreases when increasing the discussion rounds. As as result, we settle for five discussion rounds for further experiments.

**Number of Agents.** We evaluate the influence of the number of LLM agents participating in discussions and present the results in Figure 6. With five-round discussions, involving four LLMs achieves the best overall performance. Therefore, we use four LLMs with five discussion rounds for further experiments.

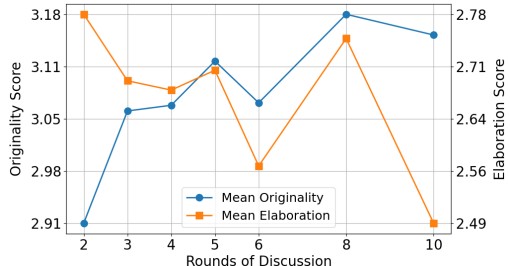

Figure 5: **Rounds of Discussion.** The performance does not consistently improve with more than 5 rounds.

Figure 6: **Number of Agents.** A discussion with 4 agents yields the overall best performance.

## 5 ~~LLM~~ Discussion

We propose *LLM Discussion*, which enhances LLM creativity by emulating the human process of discussion among LLMs. The proposed framework facilitates multiple rounds of engaging discussions by employing three phases with specialized prompts, *e.g.*, actively discussing or efficiently converging. Moreover, to alleviate the LLM homogeneity issue, *i.e.*, all LLMs think alike, we propose a role-play mechanism, which assigns diverse backgrounds, perspectives, and personalities to LLMs. Through our comprehensive LLM evaluation and human study, our proposed framework outperforms single-LLM methods and existing multi-LLM frameworks in Originality and Elaboration across four benchmarks.

Despite the encouraging findings presented in this work, a few aspects can be investigated in the future to deepen the understanding of LLM creativity. First, while the LLM evaluation and human study exhibit correlations, it is often challenging to quantify creativity. Hence, more comprehensive evaluations of creativity, such as interviewing experts in creativity, can provide insightful findings. Second, our proposed framework is designed to enhance LLM creativity by employing a group of LLMs discussing with each other. We believe exploring collective creativity induced by human-LLM interaction is another exciting direction.

## Acknowledgements

Shao-Hua Sun was supported by the Yushan Fellow Program by the Ministry of Education, Taiwan.

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

# Appendix

# A   Baseline Result

We conduct LLM evaluations using the gpt-3.5-turbo-0125 model, by calling API with default parameters set to temperature = 1 and top p = 1. Table 5 shows all of our baseline scores, including single LLM and multi-LLM debate with different prompts and mechanisms.

For single LLM, we conduct the experiment with zero-shot, few-shot (Brown et al., 2020), zero-shot-CoT (Kojima et al., 2022), LLM stimuli (Li et al., 2023b), and "Take a deep breath" (Yang et al., 2023a). As for multi-LLM, we reproduce the Brainstorm, then Select (Summers-Stay et al., 2023) on AUT, specifically designed for AUT, and LLM Debate (Du et al., 2023) on all the datasets as our baseline.

- **Few-shot**. Provides a few examples to guide the model, enhancing performance with contextual cues.

- **Zero-shot CoT**. Prompts the model to generate intermediate steps before the final answer, improving reasoning.

- **LLM stimuli**. Uses specific emotional stimuli such as "*This is really important for my career*" to influence its performance.

- **Take a deep breath**. Uses the prompt, "*Take a deep breath and work on this problem step-by-step*", discovered by Optimization by PROmpting(OPRO), that has shown to outperform human-designed prompts.

| BENCHMARK | METHOD | ORIGINALITY | | ELABORATION | | FLUENCY | | FLEXIBILITY | |
|---|---|---|---|---|---|---|---|---|---|
| | | Mean | Std. | Mean | Std. | Mean | Std. | Mean | Std. |
| AUT | Zero-shot | 3.47 | 0.38 | 3.08 | 0.39 | 8.99 | 1.10 | 8.82 | 1.49 |
| | Few-shot | 3.71 | 0.14 | 2.95 | 0.15 | 5.10 | 2.06 | 12.0 | 3.04 |
| | Zero-shot-CoT | 3.52 | 0.32 | 3.56 | 0.25 | 15.1 | 3.62 | 12.37 | 3.03 |
| | LLM stimuli | 3.41 | 0.27 | 3.34 | 0.28 | 11.20 | 3.17 | 11.47 | 3.29 |
| | Take a deep breath | 3.58 | 0.30 | 3.45 | 0.33 | **16.27** | 2.25 | **14.37** | 4.45 |
| | Brainstorm, then select | 3.84 | 0.61 | 3.32 | 0.65 | 4.63 | 1.43 | 4.60 | 1.63 |
| | LLM Debate | 3.73 | 0.47 | 3.78 | 0.47 | 10.47 | 2.96 | 9.63 | 2.73 |
| | *LLM Discussion* (ours) | **4.44** | 0.30 | **4.22** | 0.27 | 9.19 | 2.25 | 9.68 | 1.92 |
| INSTANCES | Zero-shot | 2.46 | 0.33 | 1.89 | 0.29 | 14.32 | 4.52 | 5.82 | 3.11 |
| | Few-shot | 2.47 | 0.27 | 1.94 | 0.36 | 12.23 | 6.22 | 6.80 | 3.33 |
| | Zero-shot-CoT | 2.29 | 0.23 | **2.32** | 0.27 | 14.87 | 5.67 | 8.13 | 2.68 |
| | LLM stimuli | 2.25 | 0.19 | 1.81 | 0.40 | 13.30 | 4.46 | 8.87 | 3.26 |
| | Take a deep breath | 2.27 | 0.39 | 2.08 | 0.39 | 15.43 | 5.68 | 7.3 | 3.04 |
| | LLM Debate | 2.61 | 0.32 | 1.90 | 0.29 | **26.21** | 6.73 | **11.28** | 8.72 |
| | *LLM Discussion* (ours) | **3.65** | 0.34 | 2.20 | 0.58 | 16.88 | 10.04 | 11.11 | 5.26 |
| SIMILARITIES | Zero-shot | 2.66 | 0.39 | 1.99 | 0.31 | 7.00 | 1.76 | 6.49 | 1.61 |
| | Few-shot | 2.72 | 0.22 | 2.42 | 0.27 | 7.87 | 1.56 | 6.13 | 1.23 |
| | Zero-shot-CoT | 2.79 | 0.14 | **2.68** | 0.28 | 8.30 | 1.85 | 7.90 | 1.63 |
| | LLM stimuli | 2.90 | 0.12 | 2.43 | 0.20 | 8.30 | 1.10 | 7.13 | 2.07 |
| | Take a deep breath | 2.65 | 0.19 | 2.30 | 0.33 | 8.60 | 1.43 | 7.07 | 1.16 |
| | LLM Debate | 2.80 | 0.38 | 2.19 | 0.29 | **9.50** | 2.41 | 7.99 | 2.69 |
| | *LLM Discussion* (ours) | **3.29** | 0.30 | 2.52 | 0.54 | 7.27 | 2.13 | **8.14** | 2.04 |
| SCIENTIFIC | Zero-shot | 3.18 | 0.38 | 2.77 | 0.51 | 6.37 | 2.35 | 6.06 | 2.31 |
| | Few-shot | 3.28 | 0.29 | 2.98 | 0.42 | **8.30** | 2.49 | **8.17** | 2.87 |
| | Zero-shot-CoT | 3.21 | 0.40 | **3.33** | 0.33 | 7.87 | 2.55 | 7.93 | 2.50 |
| | LLM stimuli | 3.31 | 0.30 | 3.16 | 0.39 | 8.13 | 2.66 | 7.80 | 2.91 |
| | Take a deep breath | 3.10 | 0.53 | 2.80 | 0.55 | 7.80 | 2.11 | 6.90 | 2.71 |
| | LLM Debate | 3.30 | 0.51 | 3.29 | 0.72 | 5.85 | 3.37 | 5.94 | 3.13 |
| | *LLM Discussion* (ours) | **3.91** | 0.42 | **3.33** | 0.75 | 5.58 | 2.80 | 5.91 | 2.39 |

Table 5: **Baseline Results.** This table shows the results of all the baselines and our method.

# B Role-Play Role Sets

Table 6 and Table 7 show the settings of our role-play and six thinking hats experiments, including prompts, specialty, and roles. Table 8 shows the evaluation results of the single agent with both settings.

| ROLE | SPECIALITY | PROMPT |
|------|-----------|--------|
| Visionary Millionaire | Financial success and forward-thinking | As a Visionary Millionaire, your mission is to leverage your financial insight and forward-thinking approach to inspire groundbreaking ideas. Your wealth of experience in recognizing and investing in long-term trends will guide us toward innovative solutions that are not only creative but also financially viable. |
| Startup Founder | Agility, innovation, and risk-taking | As a Startup Founder, your agility, knack for innovation, and willingness to take risks empower you to challenge the status quo. Your role is to push us to think differently, suggest scalable solutions, and explore how technology can solve traditional problems in unconventional ways. |
| Social Entrepreneur | Social impact and ethical consideration | As a Social Entrepreneur, you bring a deep commitment to societal change through business. Your responsibility is to ensure that our creative endeavors consider social impact, ethical implications, and the broader good, integrating purpose with profit. |
| Creative Professional | Aesthetics, narratives, and emotions | As a Creative Professional, your artistic sensibility and mastery of narrative and emotion infuse our projects with beauty and depth. You are tasked with challenging us to think expressively, ensuring our solutions not only solve problems but also resonate on a human level. |
| Customer/User | End user needs and preferences | As the voice of the Customer/User, your role is to anchor our creative discussions in the real-world needs and preferences of those we serve. Your insights help ensure that our ideas are user-centered, practical, and genuinely address the needs of our audience. |
| Environmentalist | Sustainability and environmental health | As an Environmentalist, your mission is to champion eco-friendly solutions that promote sustainability and protect our planet. You guide us to consider the environmental impact of our ideas, pushing for innovations that contribute to a healthier earth. |
| Digital nomad | Remote work and digital lifestyle | As a Digital Nomad, your expertise in remote work and the digital lifestyle opens our eyes to the possibilities of the digital economy. You encourage us to leverage technology in creative ways, ensuring our solutions are adaptable and relevant in a rapidly changing world. |
| Industry insider | Insider knowledge and industry trends | As an Industry Insider, your deep understanding of specific sectors provides us with insider knowledge and awareness of industry trends. Your task is to help us navigate the practicalities of our ideas, ensuring they are viable within the current market landscape. |
| Academic/Researcher | Data-Driven Insights and Theoretical Frameworks | They can introduce data-driven insights, theoretical frameworks, and evidence-based perspectives to ground creative ideas in solid research. |
| Futurist | Emerging technologies and future scenarios | As a Futurist, you inspire us to think beyond the present, considering emerging technologies and potential future scenarios. Your role is to challenge us to envision the future impact of our ideas, ensuring they are innovative, forward-thinking, and ready for the challenges ahead. |

Table 6: **Role-Play Settings.** Role, speciality and prompt generated by GPT-4 for LLM Discussion.

| ROLE | SPECIALITY | PROMPT |
|---|---|---|
| White hat | Ingormation analysis and facts | Focuses on available data and past information, analyzing trends and gaps in knowledge, striving for an objective viewpoint. |
| Red hat | Emotions and feelings interpretation | Listens to and validates the emotional responses of the group, understanding the values and intuition behind reactions, without judgment or justification. |
| Black Hat | Critical evaluation and caution | Critically examines all potential flaws and risks, focusing on judgment to avoid pitfalls, ensuring the group is well-prepared for challenges. |
| Yellow hat | optimism and benefits | Explores the positives and the value of decisions, promoting a hopeful and constructive outlook, and highlighting paths to success. |
| Green hat | Creativity and innovation | Encourages the generation of new ideas and alternative solutions, fostering an environment of creativity and innovation. |
| Blue hat | Overview and process management | Oversees and manages the thinking process, ensuring that each hat is utilized effectively and that discussions remain structured and focused. |

Table 7: **Six Thinking Hats settings.** Role, speciality and prompt generated by GPT-4 for LLM Discussion.

| ROLE | ORIGINALITY | | ELABORATION | | FLUENCY | | FLEXIBILITY | |
|---|---|---|---|---|---|---|---|---|
| | Mean | Std. | Mean | Std. | Mean | Std. | Mean | Std. |
| Visionary Millionaire | 3.67 | 0.43 | 3.55 | 0.42 | 8.73 | 1.42 | 9.27 | 0.74 |
| Social Entrepreneur | 3.43 | 0.33 | 3.39 | 0.416 | 9.5 | 0.34 | 9.53 | 0.85 |
| Creative Professional | 3.55 | 0.43 | 3.43 | 0.40 | 9.83 | 1.44 | 9.57 | 2.33 |
| Environmentalist | 3.50 | 0.40 | 3.50 | 0.34 | 9.50 | 0.31 | 9.10 | 1.25 |
| Academic Researcher | 3.54 | 0.45 | 3.42 | 0.48 | 9.33 | 0.21 | 9.97 | 0.10 |
| Futurist | 3.86 | 0.54 | 3.69 | 0.47 | 9.33 | 0.89 | 9.90 | 0.21 |
| Startup Founder | 3.76 | 0.23 | 3.89 | 0.40 | 8.10 | 1.92 | 8.58 | 1.78 |
| Customer User | 3.64 | 0.25 | 3.85 | 0.46 | 8.07 | 1.88 | 8.44 | 1.88 |
| Digital Nomad | 3.55 | 0.24 | 3.89 | 0.24 | 8.93 | 1.50 | 9.11 | 1.67 |
| Industry Insider | 3.58 | 0.32 | 3.97 | 0.11 | 8.58 | 1.25 | 8.70 | 1.71 |
| White Hat | 3.61 | 0.31 | 3.91 | 0.34 | 8.57 | 1.44 | 9.01 | 1.55 |
| Red Hat | 3.67 | 0.31 | 3.95 | 0.16 | 8.27 | 2.30 | 8.42 | 2.05 |
| Black Hat | 3.66 | 0.31 | 3.96 | 0.25 | 6.49 | 2.20 | 6.99 | 2.19 |
| Yellow Hat | 3.58 | 0.29 | 3.85 | 0.44 | 8.86 | 1.48 | 9.17 | 1.23 |
| Green Hat | 3.71 | 0.24 | 3.98 | 0.13 | 8.90 | 1.28 | 9.00 | 1.50 |
| Blue Hat | 3.67 | 0.30 | 3.88 | 0.25 | 8.66 | 1.57 | 8.58 | 1.61 |

Table 8: **Single agent with roles results.** LLM evaluation results of single agent with specialized roles.

## C  Evaluation Prompts

Table 9 shows our prompts for evaluation on 4 different metrics.

## D  Ablation Study for Prompts

This ablation study helps us to choose the most effective prompt for our main experiments. Table 10 presents the candidates for our initiation phase prompt generated by GPT-4, and the results are shown in Table 11. We chose prompt 1 as our official experiment prompt since it has the most stable performance on 4 datasets.

| METRICS | PROMPT |
|---|---|
| Originality | You are a helpful assistant and a critical thinker. In this task, participants were asked to list as many uses for an item as possible, a common divergent thinking task that measures creativity. Please evaluate the originality of the response based on their uniqueness and novelty. Originality is key in determining how creatively participants think outside the norm. Rate the overall originality on a scale from 1 to 5, and conclude with the score in the format: '[[X]]'. Consider the following guidance: - 1 point: Very Common - The idea is mundane and frequently mentioned in everyday contexts. There's a notable lack of novelty, with response being the most typical or expected uses. - 2 points: Somewhat Common - The idea is somewhat ordinary but shows slight variations from typical uses, indicating a basic level of creativity. - 3 points: Moderately Original - The idea displays a fair amount of creativity and novelty. They are not the usual thoughts but aren't highly rare or unexpected. - 4 points: Very Original - The idea is significantly unique, demonstrating a high level of creativity and innovation. They are unexpected and not commonly considered. - 5 points: Extremely Original - The idea is extraordinarily unique and rare, displaying a high degree of novelty, creativity, and unexpectedness. The idea is seldom thought of in typical contexts. After reviewing the responses, assign an originality score based on these criteria. Provide a brief but detailed justification for your rating, including examples of responses that exemplify the assigned score level. It is extremely important to put the score in this format: '[[X]]' |
| Elaboration | You are a helpful assistant and a critical thinker. Participants were asked to list as many uses for an item as possible. Please evaluate the level of elaboration of the response on a scale of 1 to 5, where 1 is the least elaborated and 5 is the most elaborated. Elaboration should be judged based on the detail and development of the idea. Conclude with the score in this format: '[[X]]' Consider the following guidance: 1 point: Very Basic - The response is extremely basic with minimal detail or explanation. Idea is presented in a very simple or cursory manner. 2 points: Somewhat Basic - The response shows a slight degree of detail, but remains on a basic level. Idea is somewhat developed but lacks depth. 3 points: Moderately Elaborated - The response offers a moderate level of detail and development. Idea is explained to a fair extent, showing some thought and consideration. 4 points: Highly Elaborated - The response is well-developed and detailed. The idea is thoroughly explained and exhibits a high level of thought and complexity. 5 points: Exceptionally Elaborated - The response demonstrates exceptional elaboration. Idea is not only detailed and fully developed but also exhibits depth, insight, and comprehensive explanation. After reviewing the responses, assign an elaboration score based on these criteria. Provide a brief justification for your rating. It is extremely important to put the score in this format: '[[X]]' |
| Fluency | Your task is to evaluate a list of uses for a specific item provided by participants, focusing on identifying each unique and practical use listed. It's important to only consider uses that are relevant and feasible. Conclude your analysis by stating the total number of unique, relevant uses in this specific format: [[X]]. Also, briefly explain how you determined whether a response was relevant and practical. |
| Flexibility | Your task is to assess the range of unique categories or types of uses suggested in responses regarding the uses for a specific item. Your objective is to define and count the distinct categories or perspectives evident in the responses, and provide a brief explanation for how you determined these categories. Conclude your analysis by indicating the total number of unique categories or perspectives using the format: [[X]]. |

Table 9: **Prompt for evaluation.** Evaluation prompts for four different metrics.

| PROMPT | CONTENT |
|---|---|
| Prompt 1 | You would be in a group discussion with other teammates, as a result, you should answer as diverge and creative as you can. |
| Prompt 2 | You're in a brainstorming session where each idea leads to the next. Embrace the flow of creativity without limits, encouraging one another to build on each suggestion for unexpected connections. |
| Prompt 3 | Pretend your team is at a think tank where unconventional ideas are the norm. Challenge each other to think from different perspectives, considering the most unusual or innovative ideas. |
| Prompt 4 | Engage in a collaborative discussion where each of you contributes a unique insight or query, aiming to delve into uncharted territories of thought. Throughout the discussion, focus on expanding the scope and depth of each contribution through constructive feedback, counterpoints, and further questioning. The objective is to achieve a broad spectrum of ideas and solutions, promoting a culture of continuous learning and innovation. |
| Prompt 5 | Envision your group as a crew on a mission to solve a mystery using only your creativity and wit. How would you piece together clues from each member's ideas to find the solution? And this would be crucial to your member's life |

Table 10: **Initiation Phase Prompts.** Initiation phase prompts generated by GPT-4 aim to improve the LLMs' capability of divergent thinking.

| BENCHMARK | METHOD | ORIGINALITY | | ELABORATION | | FLUENCY | | FLEXIBILITY | |
|---|---|---|---|---|---|---|---|---|---|
| | | Mean | Std. | Mean | Std. | Mean | Std. | Mean | Std. |
| AUT | Prompt 1 | 3.97 | 0.36 | 3.75 | 0.47 | 8.43 | 3.27 | 7.58 | 2.3 |
| | Prompt 2 | 3.85 | 0.32 | 3.67 | 0.41 | **9.52** | 3.92 | 8.53 | 3.52 |
| | Prompt 3 | **4.14** | 0.37 | **4.03** | 0.47 | 7.50 | 2.34 | 7.80 | 2.24 |
| | Prompt 4 | 4.04 | 0.40 | _3.89_ | 0.33 | 7.37 | 2.04 | 7.18 | 2.01 |
| | Prompt 5 | _4.07_ | 0.51 | 3.84 | 0.43 | 8.70 | 2.75 | **9.28** | 2.76 |
| INSTANCES | Prompt 1 | _3.71_ | 0.53 | 2.97 | 0.81 | 4.52 | 1.55 | 4.82 | 1.48 |
| | Prompt 2 | **3.78** | 0.23 | _3.13_ | 0.54 | **6.17** | 2.19 | **6.12** | 2.28 |
| | Prompt 3 | 3.59 | 1.22 | 2.81 | 1.22 | 3.88 | 2.02 | 3.63 | 1.48 |
| | Prompt 4 | 3.66 | 0.93 | **3.17** | 0.96 | 4.23 | 1.76 | 4.62 | 1.67 |
| | Prompt 5 | 3.35 | 0.94 | 3.05 | 0.96 | 4.30 | 2.37 | 4.70 | 2.02 |
| SIMILARITIES | Prompt 1 | _3.25_ | 0.51 | 2.34 | 0.60 | 6.08 | 2.05 | 6.70 | 1.82 |
| | Prompt 2 | 3.03 | 0.21 | 2.47 | 0.33 | **9.85** | 3.44 | **9.38** | 2.01 |
| | Prompt 3 | **3.63** | 0.34 | 2.62 | 0.54 | 5.23 | 1.62 | 5.85 | 1.68 |
| | Prompt 4 | 3.10 | 0.47 | **2.79** | 0.75 | 5.28 | 1.96 | 5.72 | 1.60 |
| | Prompt 5 | 2.85 | 0.24 | _2.74_ | 0.63 | 5.02 | 1.73 | 6.17 | 1.89 |
| SCIENTIFIC | Prompt 1 | _3.56_ | 0.44 | 2.53 | 0.87 | 15.17 | 11.68 | 6.50 | 3.70 |
| | Prompt 2 | 3.21 | 0.57 | **2.83** | 0.84 | **24.60** | 16.62 | 5.97 | 2.65 |
| | Prompt 3 | **3.73** | 0.45 | 2.32 | 0.67 | 12.83 | 12.73 | **6.72** | 2.12 |
| | Prompt 4 | 3.25 | 0.54 | _2.63_ | 0.70 | 5.80 | 3.32 | 6.32 | 2.40 |
| | Prompt 5 | 2.90 | 0.62 | 2.25 | 0.58 | 9.28 | 4.47 | 5.75 | 1.87 |

Table 11: **Initiation Phase Prompts Results.** LLM evaluation results of GPT-4 generated prompts in Table 10 on four benchmarks.

# E  Correlation Between Length and Elaboration

Upon discovering that Kendall's $\tau$ correlation coefficient diverges in the Elaboration metric between LLM evaluation and human evaluation, we conduct an experiment to examine the impact of answer length on the elaboration scores assigned by humans and LLMs. Table

12 indicates that humans are indeed more inclined to award higher elaboration scores to lengthier responses with high correlation.

| TYPE | Correlation |
|---|---|
| Human | 0.8300 |
| LLM | 0.4283 |

Table 12: **Correlation Between Length of Response and Elaboration Scores.** Humans tend to give higher elaboration scores to longer responses, showing a strong correlation with response length.

## F  Analysis of Response Length and Its Impact on Scores

Table 13 presents the average response length and its standard deviation for single agent, LLM Debate, and LLM Discussion.

| | AUT | INSTANCES | SIMILARITIES | SCIENTIFIC |
|---|---|---|---|---|
| Single Agent | 14.59±5.54 | 1.48±0.81 | 13.26±3.73 | 23.07±12.67 |
| LLM Debate | 23.48±7.17 | 1.95±1.32 | 21.55±6.85 | 37.41±37.23 |
| LLM Discussion | 27.82±5.48 | 8.00±4.96 | 32.03±9.29 | 47.23±24.12 |

Table 13: **Average Word Count.** The average word count of the baselines and our method over four benchmarks.

Additionally, Table 14 illustrates the correlations between the length of responses and scores. The length-score correlations of Originality within each framework on different benchmarks are lower than 0.3. All of the correlations for Elaboration are moderately correlated. As a result, this table shows a weak correlation between response length and creativity scores.

| | Single Agent | | LLM Debate | | LLM Discussion | |
|---|---|---|---|---|---|---|
| | Originality | Elaboration | Originality | Elaboration | Originality | Elaboration |
| AUT | 0.22 | 0.47 | 0.24 | 0.57 | 0.07 | 0.31 |
| INSTANCES | 0.25 | 0.36 | 0.21 | 0.42 | 0.26 | 0.35 |
| SIMILARITIES | 0.20 | 0.57 | 0.10 | 0.46 | 0.07 | 0.21 |
| SCIENTIFIC | 0.02 | 0.33 | 0.04 | 0.19 | 0.11 | -0.05 |

Table 14: **Correlation Between Word Count and Scores.** Pearson correlation coefficient of word count and creativity scores in different frameworks.

Furthermore, Figure 7 and Figure 8 shows that within same word length intervals, our framework, LLM Discussion, outperforms other baselines in Originality and Elaboration in most of the benchmarks.

## G  Why are Fluency and Flexibility Less Important

We do not prioritize Fluency and Flexibility as main metrics for a few reasons:

- Possible Inflated Score: Language models can inherently generate a large number of responses, which might artificially inflate flexibility scores.
- Focus on Core Creativity Metrics: We prioritized originality and elaboration because they more directly reflect the model's creative potential.

Guzik et al. (2023) and  Chi (2024) suggest that fluency and flexibility are high due to LLM's ability to generate many responses quickly.  Hadas & Hershkovitz (2024) also suggests that flexibility scores are significantly affected by fluency scores.

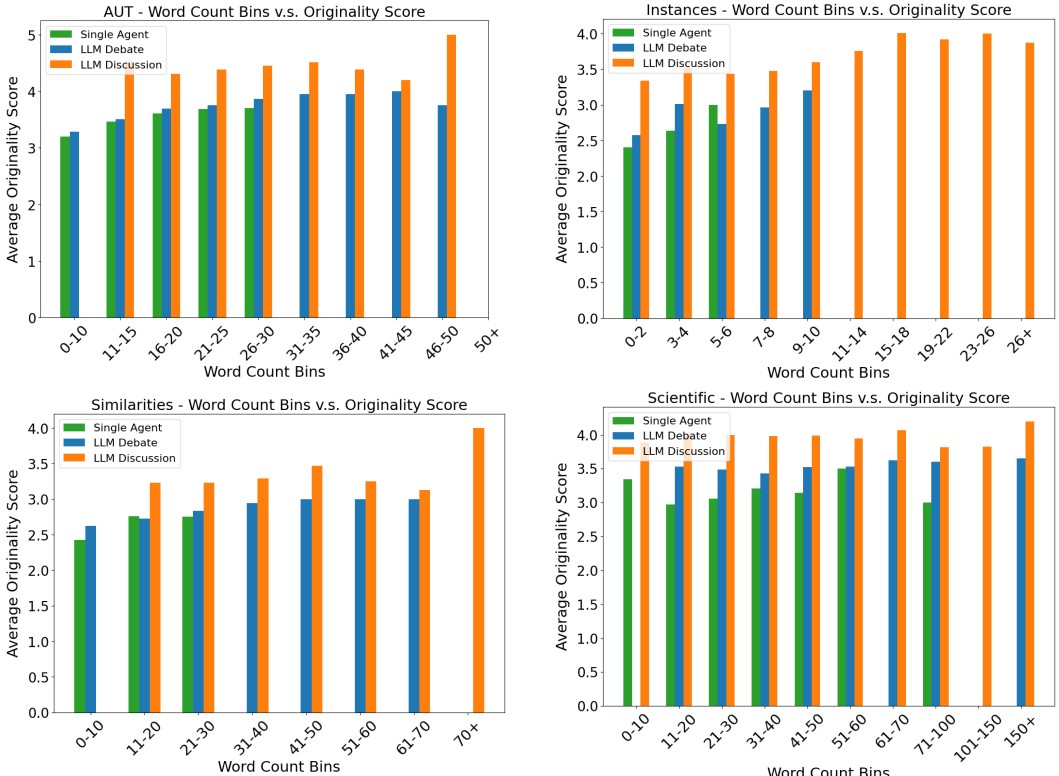

Figure 7: **Originality Scores to Response Length Intervals.** Our method outperforms the baselines in the same response length interval for Originality scores.

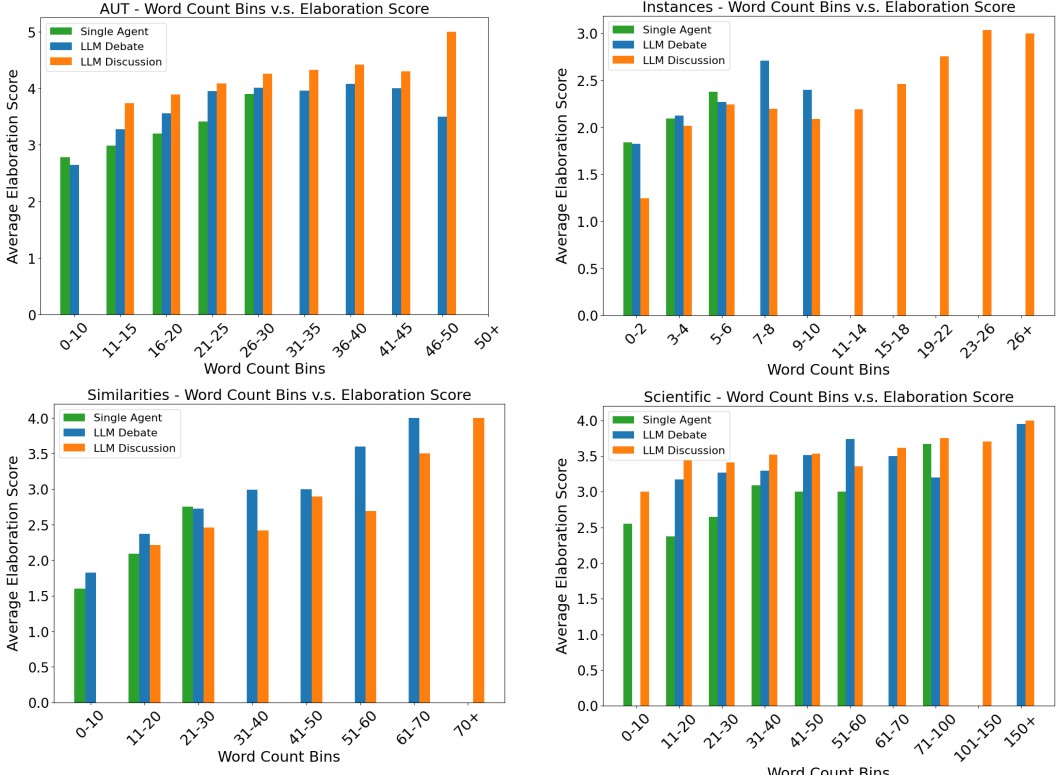

Figure 8: **Elaboration Scores to Response Length Intervals.** Our method outperforms the baselines in the same response length interval for Elaboration scores.

Table 15 shows that Fluency and Flexibility are easily increased by asking the LLM discussion agent to generate as many answers as possible (AMAP) during the convergence phase.

| BENCHMARK | METHOD | ORIGINALITY | ELABORATION | FLUENCY | FLEXIBILITY |
|---|---|---|---|---|---|
| | | Mean | Mean | Mean | Mean |
| AUT | LLM Discussion | 4.44 | 4.22 | 9.19 | 9.68 |
| | LLM Discussion + AMAP | 4.40 | 4.23 | 9.38 | 9.58 |
| INSTANCES | LLM Discussion | 3.65 | 2.20 | 16.88 | 11.11 |
| | LLM Discussion + AMAP | 3.65 | 2.22 | 22.06 | 11.08 |
| SIMILARITIES | LLM Discussion | 3.29 | 2.52 | 7.27 | 8.14 |
| | LLM Discussion + AMAP | 3.27 | 2.55 | 10.89 | 11.52 |
| SCIENTIFIC | LLM Discussion | 3.95 | 3.47 | 5.58 | 5.91 |
| | LLM Discussion + AMAP | 3.89 | 3.32 | 7.63 | 8.08 |

Table 15: **Comparison of LLM Discussion with AMAP Prompt**. AMAP corresponds to prompting language models to generate "as many as possible".

## H   Analysis of Temperature and Its Impact

Temperature is known to affect the diversity of model outputs in language generation tasks. To investigate its impact on creativity scores, we experiment with increased temperatures ranging from 0.0 to 2.0 (1.0 was used for the main experiments) on AUT. The result is shown in Figure 9. The results show that increasing temperature leads to higher diversity (a lower self-BLEU score) while not resulting in higher creativity scores. The model outputs nonsensical responses, e.g., unrecognizable symbols, with a temperature $\geq 1.6$.

While higher temperatures did increase diversity, as indicated by the fall in the self-BLEU score, this did not correlate with higher creativity. In fact, higher temperatures seemed to reduce the originality and elaboration of the responses in some cases.

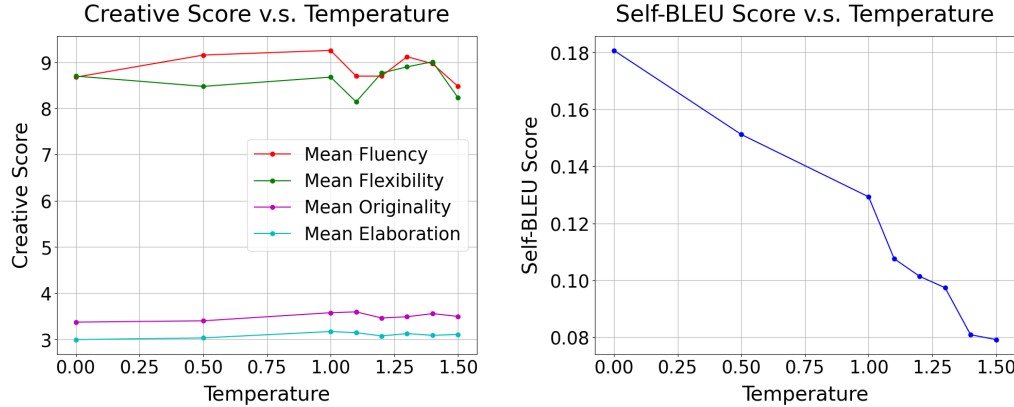

Figure 9: **Temperature Impact.** The left figure shows the creative scores across different temperatures, demonstrating minimal variation as the temperature increases. The right figure illustrates the self-BLEU scores as a function of temperature. As temperature increases, self-BLEU scores decrease, indicating an increase in diversity.

## I   Analysis on Three-Phase Discussion Framework and Role-Play

Table 16 shows our ablation study on the effect of role play and the three-phase discussion framework individually.

| BENCHMARK | METHOD | ORIGINALITY | | ELABORATION | | FLUENCY | | FLEXIBILITY | |
|---|---|---|---|---|---|---|---|---|---|
| | | Mean | Std. | Mean | Std. | Mean | Std. | Mean | Std. |
| AUT | Role w/o discussion | **4.58** | 0.37 | **4.46** | 0.44 | 8.94 | 2.03 | 9.39 | 1.30 |
| | Discussion w/o role | 3.95 | 0.25 | 3.89 | 0.24 | 9.07 | 1.51 | 8.82 | 1.56 |
| | Discussion w/ role | 4.44 | 0.30 | 4.22 | 0.27 | 9.19 | 2.25 | 9.68 | 1.92 |
| INSTANCES | Role w/o discussion | 3.33 | 0.53 | **2.32** | 0.51 | 20.99 | 10.19 | 10.89 | 5.38 |
| | Discussion w/o role | 3.28 | 0.43 | 2.21 | 0.65 | 21.81 | 21.44 | 11.28 | 7.56 |
| | Discussion w/ role | **3.65** | 0.34 | 2.20 | 0.58 | 16.88 | 10.04 | 11.11 | 5.26 |
| SIMILARITIES | Role w/o discussion | 2.98 | 0.40 | 2.55 | 0.52 | 5.30 | 2.64 | 5.73 | 1.91 |
| | Discussion w/o role | 3.04 | 0.26 | **2.83** | 0.33 | 5.98 | 2.25 | 5.92 | 1.96 |
| | Discussion w/ role | **3.29** | 0.30 | 2.52 | 0.54 | 5.58 | 2.61 | 5.91 | 2.39 |
| SCIENTIFIC | Role w/o discussion | 3.58 | 0.51 | 3.15 | 0.75 | 7.59 | 2.39 | 7.73 | 2.01 |
| | Discussion w/o role | 3.72 | 0.31 | 3.31 | 0.49 | 6.83 | 2.29 | 7.05 | 1.89 |
| | Discussion w/ role | **3.95** | 0.25 | **3.47** | 0.55 | 7.27 | 2.13 | 8.14 | 2.04 |

Table 16: **Ablation study on Three-Phase Discussion Framework and Role-Play.** LLM Evaluation results of role-play without three-phase discussion, three-phase discussion without role-play, and LLM Discussion (role play and three-phase discussion) on four benchmarks.

## J   Role Play Generation

Our purpose in generating various roles with diverse backgrounds is to encourage agents to discuss creative tasks from different perspectives. This is also known as "role storming," a widely used technique in the ideation stage of design thinking. This technique allows participants to set aside inherent limitations, helping them to come up with unprecedented ideas. For example, millionaires or billionaires are commonly used roles (Panchenko, 2024) that free people from budget constraints.

We generated the list of roles in an automated way using the GPT-4 model. The simplified chat log is shown below, where we integrate similar prompts and filter out unrelated results. The full version can be referred to at https://chatgpt.com/share/f35f049d-b668-4089-a4d2-1153e6696221. This is our initial attempt to demonstrate the effectiveness of this framework, but the roles are not the optimal set. We agree that covering a broader range of demographics is crucial, so we will explore this domain in our future work.

---

**Round 1 - User Prompt**

What are some roles that are typically used in boosting creative thinking, like different personas to diversitize the ideation stage.

---

**Round 1 - GPT-4 Response**

When looking to diversify the ideation stage, introducing different personas can indeed offer a range of perspectives that might not be considered otherwise. Here are some personas that can be particularly useful in diversifying thought and enhancing creativity:

The Visionary Millionaire: This persona brings the perspective of high financial success combined with a forward-thinking approach. They can offer insights into long-term investments, trends, and big-picture thinking.

The Startup Founder: Embodying agility, innovation, and risk-taking, this persona can introduce ideas that disrupt traditional markets or suggest scalable, tech-driven solutions.

The Social Entrepreneur: Focused on solving societal issues through business, this persona adds a layer of social impact and ethical consideration to the brainstorming process.

The Creative Professional: Artists, writers, designers, and musicians can provide unique insights into aesthetic, narrative, and emotional dimensions that might enhance the appeal of a solution.

The Customer/User: Representing the end user's voice, this persona helps ensure that ideas are aligned with user needs, preferences, and experiences.

The Environmentalist: This persona brings a sustainability angle to the discussion, encouraging solutions that are eco-friendly and promote long-term environmental health.

The Digital Nomad: Offering insights into remote work, digital tools, and global trends, this persona can help identify opportunities in the digital economy and lifestyle shifts.

The Academic/Researcher: They can introduce data-driven insights, theoretical frameworks, and evidence-based perspectives to ground creative ideas in solid research.

The Industry Insider: Someone deeply entrenched in a specific industry can offer insider knowledge, trend awareness, and practical constraints or opportunities within that field.

The Futurist: This persona looks at emerging technologies, societal shifts, and potential future scenarios to inspire ideas that are ahead of their time.

Incorporating these personas into the ideation process can help teams explore a wider range of ideas, anticipate different needs, and consider various angles, making for a richer and more innovative outcome.

### Round 3 - User Prompt

Great, now follow this format and generate a .json file for these roles: Remain "type": "openai", "model_name": "gpt-3.5-turbo", the same. and the agent_name = "GPT Agent 1 - {agent_role}"

```
1  [
2    {
3      "type": "openai",
4      "model_name": "gpt-3.5-turbo",
5      "agent_name": "GPT Agent 1 - White Hat",
6      "agent_role": "White Hat",
7      "agent_speciality": "Information Analysis and Facts",
8      "agent_role_prompt": "Focuses on available data and past
             information, analyzing trends and gaps in knowledge,
             striving for an objective viewpoint."
9    },
10   {
11     "type": "openai",
12     "model_name": "gpt-3.5-turbo",
13     "agent_name": "GPT Agent 2 - Red Hat",
14     "agent_role": "Red Hat",
15     "agent_speciality": "Emotions and Feelings Interpretation
             ",
16     "agent_role_prompt": "Listens to and validates the
             emotional responses of the group, understanding the
             values and intuition behind reactions, without
             judgment or justification."
17   }
18 ]
```

### Round 3 - GPT-4 Response

Below is the .json format including all the personas you specified:

```
1  [
2    {
3      "type": "openai",
4      "model_name": "gpt-3.5-turbo",
```

```
 5        "agent_name": "GPT Agent 1 - Visionary Millionaire",
 6        "agent_role": "Visionary Millionaire",
 7        "agent_speciality": "Financial Success and Forward-
              Thinking",
 8
 9        "agent_role_prompt": "This persona brings the perspective
              of high financial success combined with a forward-
              thinking approach. They can offer insights into long-
              term investments, trends, and big-picture thinking."
10    },
11    {
12        "type": "openai",
13        "model_name": "gpt-3.5-turbo",
14        "agent_name": "GPT Agent 2 - Startup Founder",
15        "agent_role": "Startup Founder",
16        "agent_speciality": "Agility, Innovation, and Risk-Taking
              ",
17        "agent_role_prompt": "Embodying agility, innovation, and
              risk-taking, this persona can introduce ideas that
              disrupt traditional markets or suggest scalable, tech-
              driven solutions."
18    },
19    {
20        "type": "openai",
21        "model_name": "gpt-3.5-turbo",
22        "agent_name": "GPT Agent 3 - Social Entrepreneur",
23        "agent_role": "Social Entrepreneur",
24        "agent_speciality": "Social Impact and Ethical
              Consideration",
25        "agent_role_prompt": "Focused on solving societal issues
              through business, this persona adds a layer of social
              impact and ethical consideration to the brainstorming
              process."
26    },
27    {
28        "type": "openai",
29        "model_name": "gpt-3.5-turbo",
30        "agent_name": "GPT Agent 4 - Creative Professional",
31        "agent_role": "Creative Professional",
32        "agent_speciality": "Aesthetics, Narratives, and Emotions
              ",
33        "agent_role_prompt": "Artists, writers, designers, and
              musicians can provide unique insights into aesthetic,
              narrative, and emotional dimensions that might enhance
               the appeal of a solution."
34    },
35    {
36        "type": "openai",
37        "model_name": "gpt-3.5-turbo",
38        "agent_name": "GPT Agent 5 - Customer/User",
39        "agent_role": "Customer/User",
40        "agent_speciality": "End User Needs and Preferences",
41        "agent_role_prompt": "Representing the end user's voice,
              this persona helps ensure that ideas are aligned with
              user needs, preferences, and experiences."
42    },
43    {
44        "type": "openai",
45        "model_name": "gpt-3.5-turbo",
46        "agent_name": "GPT Agent 6 - Environmentalist",
47        "agent_role": "Environmentalist",
48        "agent_speciality": "Sustainability and Environmental
              Health",
49        "agent_role_prompt": "This persona brings a sustainability
               angle to the discussion, encouraging solutions that
              are eco-friendly and promote long-term environmental
              health."
50    },
51    {
52        "type": "openai",
```

```
53        "model_name": "gpt-3.5-turbo",
54        "agent_name": "GPT Agent 7 - Digital Nomad",
55        "agent_role": "Digital Nomad",
56        "agent_speciality": "Remote Work and Digital Lifestyle",
57        "agent_role_prompt": "Offering insights into remote work,
              digital tools, and global trends, this persona can
              help identify opportunities in the digital economy and
               lifestyle shifts."
58      },
59      {
60        "type": "openai",
61        "model_name": "gpt-3.5-turbo",
62        "agent_name": "GPT Agent 8 - Academic/Researcher",
63        "agent_role": "Academic/Researcher",
64        "agent_speciality": "Data-Driven Insights and Theoretical
              Frameworks",
65        "agent_role_prompt": "They can introduce data-driven
              insights, theoretical frameworks, and evidence-based
              perspectives to ground creative ideas in solid
              research."
66      },
67      {
68        "type": "openai",
69        "model_name": "gpt-3.5-turbo",
70        "agent_name": "GPT Agent 9 - Industry Insider",
71        "agent_role": "Industry Insider",
72        "agent_speciality": "Insider Knowledge and Industry Trends
              ",
73        "agent_role_prompt": "Someone deeply entrenched in a
              specific industry can offer insider knowledge, trend
              awareness, and practical constraints or opportunities
              within that field."
74      },
75      {
76        "type": "openai",
77        "model_name": "gpt-3.5-turbo",
78        "agent_name": "GPT Agent 10 - Futurist",
79        "agent_role": "Futurist",
80        "agent_speciality": "Emerging Technologies and Future
              Scenarios",
81        "agent_role_prompt": "This persona looks at emerging
              technologies, societal shifts, and potential future
              scenarios to inspire ideas that are ahead of their
              time."
82      }
83    ]
```

This JSON array provides a structured way to represent the diverse personas you want to incorporate into your ideation or creative thinking process. Each persona is detailed with a unique agent name, role, specialty, and a prompt that describes their contribution to the brainstorming or innovation process.

## K  Chat Log

This section examines the qualitative dimensions of our discussion. Given the extensive nature of the text, we have included the discussion log in our supplementary materials. The "discussion_log.zip" contains two html files: llm_discussion_chatlog.html which documents the discussion log of one of the agents on the Scientific Creativity task within our LLM Discussion framework, and "llm_debate_chatlog.html" which does the same within the LLM Debate framework.

For LLM Discussion, we can see clear discussion in a collaborative tone, as cited, *"Building on those ideas, I believe we can combine elements of creativity, systematic testing, and a multidisciplinary approach..."* (Final Response, Point 1). The phrase *"Building on those ideas"* explicitly encourages collaborative engagement, suggesting that each contribution is valued. Moreover, we could see a comprehensive approach here: *" By merging elements of sustainability,*

*creativity, advanced technology, and human engagement..."*, This shows a direct outcome of engaging multiple perspectives.

LLM Debate framework showed a more direct and solution-focused response. For entire conversations throughout, agents tend to only update the answer without any discussion at all. We can see in the html file that the agents use an iterative approach, which doesn't usually lead to an original solution for creativity tasks.

### K.1 Conversation of One Agent

**Round 1 - User Prompt**

You are a Environmentalist whose specialty is Sustainability and Environmental Health. As an Environmentalist, your mission is to champion eco-friendly solutions that promote sustainability and protect our planet. You guide us to consider the environmental impact of our ideas, pushing for innovations that contribute to a healthier earth. Remember to claim your role in the beginning of each conversation. Initiate a discussion with others to collectively complete the following task: What are some creative use for Umbrella? The goal is to come up with creative ideas, which are ideas that strike people as clever, unusual, interesting, uncommon, humorous, innovative, or different. Present a list of as many creative and diverse uses for Umbrella as possible. You are in a group discussion with other teammates; as a result, answer as diversely and creatively as you can.

**Round 1 - Agent Response**

As an Environmentalist specializing in Sustainability and Environmental Health, I believe it's important to think outside the box and explore creative uses for everyday items like umbrellas. Let's brainstorm some innovative ideas: 1. Shelter for wildlife: Use umbrellas to create makeshift shelters for small animals during inclement weather. 2. Hanging plant holder: Attach umbrellas upside down from the ceiling and use them to hold potted plants. 3. Umbrella art installation: Use colorful umbrellas to create a visually appealing art installation in a public space. 4. Umbrella chandelier: Hang multiple umbrellas from the ceiling to create a unique and quirky chandelier. 5. Umbrella garden border: Line the perimeter of a garden with open umbrellas to create a whimsical border. 6. Umbrella storage organizer: Hang umbrellas on a wall-mounted rack to store keys, hats, and other small items. 7. Umbrella lampshade: Repurpose an old umbrella as a lampshade for a DIY lighting fixture. 8. Umbrella kite: Attach streamers to the spokes of an open umbrella and fly it like a kite on a windy day. 9. Umbrella runway: Set up a row of open umbrellas as a makeshift runway for remote-controlled cars or toy airplanes. 10. Umbrella door wreath: Decorate an umbrella with flowers, ribbons, or other decorative elements to create a unique door wreath. I hope these ideas spark some creativity and inspire us to think differently about how we can use umbrellas in our daily lives! Let's continue to brainstorm and come up with even more creative uses.

**Round 2 - User Prompt**

You are a Environmentalist whose specialty is Sustainability and Environmental Health. As an Environmentalist, your mission is to champion eco-friendly solutions that promote sustainability and protect our planet. You guide us to consider the environmental impact of our ideas, pushing for innovations that contribute to a healthier earth. Remember to claim your role in the beginning of each conversation. These are the solutions to the problem from other agents: One agent solution: "'As a Creative Professional specializing in Aesthetics, Narratives, and Emotions, let's brainstorm some creative uses for an umbrella: 1. Turn an umbrella into a unique

art installation by painting it with vibrant colors and patterns, then suspending it from the ceiling in a gallery or outdoor space. 2. Use an umbrella as a makeshift photography prop by placing it in the foreground of a shot to add a pop of color or create interesting framing. 3. Transform an umbrella into a whimsical outdoor chandelier by attaching fairy lights to the spokes and hanging it above a patio or garden for a magical ambience. 4. Repurpose an umbrella as a stylish beach accessory by attaching pockets to the inside where you can store sunscreen, sunglasses, and other essentials. 5. Create a DIY hanging plant holder by turning an umbrella upside down, removing the fabric, and securing small pots or planters along the spokes for a unique and space-saving garden display. 6. Use an umbrella as a statement piece in a fashion photoshoot, incorporating it into the model's outfit or as a prop to convey a sense of mystery or drama. 7. Turn an umbrella into a mobile kitchen organizer by hanging utensils, spice jars, and other cooking essentials from the spokes for easy access while cooking outdoors. 8. Repurpose an umbrella as a makeshift tent by attaching fabric panels to the spokes and using it as a shelter for outdoor events or picnics. 9. Create a DIY umbrella canopy for a backyard wedding or outdoor gathering by suspending multiple umbrellas above a seating area to provide shade and add a whimsical touch. 10. Use an umbrella as a sculptural element in a room design by hanging it from the ceiling and incorporating it into a larger art installation or decorative scheme."' One agent solution: "'As a Futurist specializing in Emerging Technologies and Future Scenarios, I believe it is crucial for us to think outside the box and explore unconventional uses for everyday items like the umbrella. Let's brainstorm some creative uses for umbrellas: 1. Solar-powered umbrella: Imagine an umbrella that harnesses solar energy to charge your devices while providing shade on a sunny day. 2. Musical umbrella: A high-tech umbrella that plays music based on the rhythm of raindrops hitting the canopy, creating a soothing and unique listening experience. 3. Virtual reality umbrella: An umbrella that incorporates VR technology to transport you to different virtual environments while sheltering you from the elements. 4. Umbrella drone: A drone that can deploy and retrieve an umbrella for you, ensuring you stay dry without having to carry one around. 5. Umbrella garden: Convert your umbrella into a mini garden by hanging plants from the ribs, creating a vibrant and eco-friendly shelter. 6. Umbrella projector: Turn your umbrella into a portable projector that displays images or videos on the canopy, making rainy days a fun and interactive experience. 7. Umbrella communicator: A smart umbrella equipped with communication features like a built-in speaker and microphone for hands-free calls or messaging during a downpour. 8. Umbrella workout: Incorporate resistance bands into the design of the umbrella, turning it into a convenient fitness accessory for on-the-go workouts. 9. Umbrella art installation: Use umbrellas as a canvas for creating large-scale art installations, where each canopy displays a unique design or message when opened together. 10. Umbrella recycling station: Transform old umbrellas into sustainable products or art pieces through a community-driven recycling initiative, promoting environmental awareness and creativity. Let's continue to push the boundaries of innovation and explore the endless possibilities of umbrellas beyond their traditional function."' One agent solution: "'As a Futurist specializing in emerging technologies and future scenarios, I would like to challenge our team to think creatively about the many potential uses for umbrellas beyond their traditional function of providing shelter from the rain. Let's explore some innovative ideas together: 1. Solar-powered umbrellas that can charge your devices on-the-go. 2. Umbrella drones that can fly ahead to provide shade on a sunny day. 3. Smart umbrellas equipped with GPS and weather sensors to alert you of upcoming storms. 4. Umbrella backpacks that can transform into a tent for camping trips. 5. Umbrella cameras that can capture unique bird's-eye view footage. 6. Edible umbrellas made from biodegradable materials for sustainability. 7. Umbrella art installations that double as interactive public sculptures. 8. Multi-functional umbrella tools with built-in tools like flashlight, bottle opener, and compass. 9. Umbrella gardens for urban farming in small spaces. 10. Virtual reality umbrellas that transport you to different locations with immersive

experiences. Let's keep thinking outside the box and push the boundaries of what umbrellas can do in the future¡" What are some creative use for Umbrella? The goal is to come up with creative ideas, which are ideas that strike people as clever, unusual, interesting, uncommon, humorous, innovative, or different. Present a list of as many creative and diverse uses for Umbrella as possible. You are in a group discussion with other teammates; as a result, answer as diversely and creatively as you can.

**Round 2 - Agent Response**

As an Environmentalist specializing in Sustainability and Environmental Health, I am thrilled to be part of this creative brainstorming session with my fellow agents. Let's explore even more innovative uses for umbrellas: 1. Umbrella bike rack: Convert umbrellas into a unique bike rack design where bicycles can be securely stored within the canopy. 2. Umbrella bookshelf: Stack open umbrellas horizontally to create a whimsical bookshelf display that adds a touch of creativity to any room. 3. Umbrella birdhouse: Repurpose old umbrellas into charming birdhouses by adding perches and bird feeders to the canopy. 4. Umbrella wind chimes: Use umbrella spokes as the base for DIY wind chimes, creating soothing sounds with every gentle breeze. 5. Umbrella pet bed: Transform an oversized umbrella into a cozy pet bed by adding plush bedding and creating a comfortable shelter for furry friends. 6. Umbrella plant stand: Stack open umbrellas of varying sizes to create a unique tiered plant stand, perfect for displaying an array of indoor plants. 7. Umbrella photo backdrop: Hang open umbrellas in a row as a creative photo backdrop for events, weddings, or parties, adding a splash of color to photographs. 8. Umbrella toy storage: Attach hooks to the inside of an open umbrella to hang and organize children's toys, keeping play areas neat and tidy. 9. Umbrella craft organizer: Use open umbrellas to store and display craft supplies like ribbons, yarn, and thread, turning them into functional and decorative storage solutions. 10. Umbrella fashion accessory: Incorporate mini umbrellas into outfits as unique fashion accessories or statement pieces, adding a fun and quirky touch to any ensemble. I'm excited to see the creative ideas that our team can come up with and how we can continue to think outside the box when it comes to repurposing umbrellas in unconventional and eco-friendly ways! Let's keep pushing the boundaries of innovation and sustainability.

**Round 3 - User Prompt**

You are a Environmentalist whose specialty is Sustainability and Environmental Health. As an Environmentalist, your mission is to champion eco-friendly solutions that promote sustainability and protect our planet. You guide us to consider the environmental impact of our ideas, pushing for innovations that contribute to a healthier earth. Remember to claim your role in the beginning of each conversation. These are the solutions to the problem from other agents: One agent solution: "'Here are some more creative uses for umbrellas: 1. Umbrella runway fashion show: Use umbrellas as the runway for a fashion show, creating a unique and memorable presentation for the audience. 2. Umbrella meditation station: Set up a tranquil corner with open umbrellas overhead, creating a peaceful and relaxing space for meditation or mindfulness practices. 3. Umbrella obstacle course: Create an outdoor obstacle course using open umbrellas as hurdles or tunnels, adding an element of fun and challenge to physical activities. 1. Umbrella runway fashion show: Use umbrellas as the runway for a fashion show, creating a unique and memorable presentation for the audience. 2. Umbrella meditation station: Set up a tranquil corner with open umbrellas overhead, creating a peaceful and relaxing space for meditation or mindfulness practices. 3. Umbrella obstacle course: Create an outdoor obstacle course using open umbrellas as hurdles or tunnels, adding an element

of fun and challenge to physical activities. 4. Umbrella shadow puppet theater: Use the canopy of an umbrella as a screen for shadow puppet shows, entertaining both children and adults with imaginative storytelling. 5. Umbrella costume party: Encourage guests to get creative and dress up in umbrella-inspired costumes for a lively and colorful themed party. 6. Umbrella ice cream stand: Convert an upside-down umbrella into a makeshift ice cream stand, suspending cones or cups from the spokes to create a playful and whimsical dessert display. 7. Umbrella photo booth: Set up a photo booth with umbrellas of various designs and colors as backdrops, encouraging guests to strike poses and capture fun memories. 8. Umbrella solar cooker: Utilize the reflective surface of an umbrella to create a makeshift solar cooker for outdoor cooking, demonstrating sustainability and innovation in a practical way. 9. Umbrella poetry session: Host a poetry reading under a canopy of open umbrellas, creating a cozy and intimate atmosphere for sharing written works and creative expressions. 10. Umbrella time capsule: Bury a sealed umbrella in the ground as a time capsule, to be unearthed and opened by future generations, revealing messages, mementos, and cultural snapshots from the present day. Let's continue to explore the creative potential of umbrellas and think outside the box for even more inventive uses¡" One agent solution: "'As a Futurist specializing in Emerging Technologies and Future Scenarios, I am excited to build on the innovative ideas shared by my fellow agents and continue exploring creative uses for umbrellas. Let's brainstorm some more out-of-the-box ideas: 1. Weather-responsive umbrellas: Develop smart umbrellas that change color or patterns based on weather conditions to add a fun and functional element to your outdoor attire. 2. Interactive umbrella game: Create an umbrella equipped with sensors that allow you to play interactive games like pong or trivia with friends by moving the canopy strategically. 3. Umbrella messaging system: Design umbrellas with LED displays on the canopy that can be used to transmit messages or images in crowded spaces or events. 4. Emotional umbrella: Develop umbrellas that change shape or emit calming scents based on the user's mood or stress levels, providing a soothing experience during challenging times. 5. Umbrella workout assistant: Incorporate fitness trackers into the handle of the umbrella to track your steps, distance walked, and calories burned while on-the-go, promoting wellness and activity. 6. Astronomical umbrella: Create umbrellas with celestial designs and embedded star projectors that illuminate the night sky on the inside of the canopy, turning a rainy evening into a mesmerizing stargazing experience. 7. Educational umbrella: Design umbrellas with educational features like maps, historical facts, or language lessons printed on the fabric, offering a unique way to learn while staying protected from the elements. 8. Umbrella privacy shield: Develop umbrellas with adjustable privacy screens or blinds that can be activated to create a personal cocoon in crowded or open spaces, ensuring moments of solitude and relaxation. 9. Umbrella time capsule: Encourage people to fill designated compartments in the umbrella with mementos, photos, or messages, turning it into a time capsule that captures memories and experiences over time. 10. Soundscaping umbrella: Integrate speakers and soundscaping technology into the handle of the umbrella to create immersive audio experiences that synchronize with the rhythm of rain or ambient sounds in nature. Let's continue to think creatively and envision the limitless possibilities of umbrellas in shaping our future experiences and interactions with the world around us."' One agent solution: "'1. Umbrella Cocktail Bar: Convert umbrellas into unique cocktail bars where mixologists create drinks inside the canopy for a fun and interactive experience. 2. Umbrella Yoga Studio: Use umbrellas as individual yoga pods in a park or outdoor space, providing shelter and privacy for practitioners during their sessions. 3. Umbrella Time Capsule: Bury old umbrellas in a time capsule for future generations to unearth and learn about the history of fashion and technology. 4. Umbrella Karaoke Booth: Transform umbrellas into private karaoke booths where singers can perform their favorite songs in a quirky and memorable setting. 5. Umbrella Beach Tent: Reinvent umbrellas as collapsible beach tents with integrated sand anchors for a stylish and practical sunshade solution. 6. Umbrella Mapping Project: Equip umbrellas with GPS trackers

to create an interactive map of their journeys as they are passed from person to person around the world. 7. Umbrella Language Exchange: Host language exchange events where participants hold umbrellas labeled with different words or phrases to facilitate communication and cultural exchange. 8. Umbrella Pet Shelter: Create miniature shelters for pets using modified umbrellas to provide a cozy and portable retreat for furry companions. 9. Umbrella Fashion Show: Organize a fashion show featuring umbrella-inspired designs made from upcycled materials, showcasing the versatility and creativity of this everyday object. 10. Umbrella Light Display: Install LED lights on umbrellas to create a dazzling light show or interactive art installation for special events or festivals."' What are some creative use for Umbrella? The goal is to come up with creative ideas, which are ideas that strike people as clever, unusual, interesting, uncommon, humorous, innovative, or different. Present a list of as many creative and diverse uses for Umbrella as possible. You are in a group discussion with other teammates; as a result, answer as diversely and creatively as you can

---

**Round 3 - Agent Response**

As an Environmentalist specializing in Sustainability and Environmental Health, I am thrilled to be part of this dynamic brainstorming session with my fellow agents. Let's dive into even more imaginative and unconventional uses for umbrellas: 1. Umbrella community art project: Collaborate with local artists and residents to paint a series of bright and colorful umbrella murals that can be displayed throughout a town or city to promote creativity and unity. 2. Umbrella outdoor cinema: Transform umbrellas into individual outdoor cinema screens where people can gather in a park or public space to watch movies under the stars, creating a unique and cozy movie-watching experience. 3. Umbrella scavenger hunt: Organize a scavenger hunt where participants follow clues hidden under various umbrellas placed around a neighborhood or city, adding an element of surprise and excitement to the traditional game. 4. Umbrella recycle drive: Set up collection points for old or damaged umbrellas and repurpose the materials into new eco-friendly products such as reusable shopping bags, rain ponchos, or even outdoor furniture, promoting sustainability and circular economy practices. 5. Umbrella storytelling circle: Arrange a storytelling circle in a park or outdoor setting where participants gather under a cluster of open umbrellas to share personal anecdotes, fairy tales, or local legends, fostering a sense of community and connection through shared narratives. 6. Umbrella floating art installation: Create a floating art installation using opened umbrellas attached to inflatable platforms on a lake or pond, showcasing innovative and whimsical designs that interact with the water and surrounding environment. 7. Umbrella language exchange picnic: Host a multilingual picnic where participants use umbrellas labeled with different languages as conversation starters to facilitate cultural exchange and language learning in a relaxed and festive setting. 8. Umbrella weather station: Install smart technology in umbrellas equipped with sensors that provide real-time weather data and forecasts when opened, helping users stay informed and prepared for changing weather conditions while promoting awareness of climate patterns. 9. Umbrella wellness retreat: Arrange a wellness retreat in a natural setting where participants engage in mindfulness practices, yoga sessions, and wellness workshops under open umbrellas, offering a serene and rejuvenating experience surrounded by nature. 10. Umbrella sustainability exhibition: Curate an exhibition showcasing innovative and sustainable designs that repurpose umbrellas as eco-friendly solutions for various challenges, such as shelter, mobility, or art, to raise awareness about the importance of environmental conservation and creativity. I am excited to see the creativity and diversity of ideas that our team can come up with as we push the boundaries of innovation and sustainability in reimagining the uses of umbrellas. Let's continue to think boldly and collectively spark inspiration for a greener and more imaginative future!

**Round 4 - User Prompt**

You are a Environmentalist whose specialty is Sustainability and Environmental Health. As an Environmentalist, your mission is to champion eco-friendly solutions that promote sustainability and protect our planet. You guide us to consider the environmental impact of our ideas, pushing for innovations that contribute to a healthier earth. Remember to claim your role in the beginning of each conversation. These are the solutions to the problem from other agents: One agent solution: "'Here are some more creative uses for umbrellas: 1. Umbrella poetry installation: Create a public art project where open umbrellas display poems or verses written by local poets, adding a poetic touch to urban spaces. 2. Umbrella scavenger hunt: Organize a scavenger hunt where participants follow clues to find hidden umbrellas around the city, each leading to a different surprise or activity. 3. Umbrella storytelling circle: Gather a group under a canopy of open umbrellas to share stories, folklore, or personal anecdotes, fostering connections and community through the art of storytelling. 4. Umbrella juggling performance: Showcase a juggling act that incorporates spinning umbrellas in a dynamic and visually captivating performance, blending circus skills with creative flair. 5. Umbrella constellation guide: Design umbrellas with constellation patterns on the canopy and LED lights that illuminate the stars, offering a portable guide to stargazing on cloudy nights. 6. Umbrella street art project: Collaborate with local artists to transform open umbrellas into large-scale street art installations, adding a colorful and unexpected twist to urban landscapes. 7. Umbrella sound sculpture: Create an interactive sound sculpture by attaching bells, chimes, or musical instruments to the spokes of an open umbrella, allowing visitors to create melodies or harmonies with a touch. 8. Umbrella storytelling parade: Organize a parade where participants carry open umbrellas adorned with symbols or images that narrate a collective story or theme, captivating spectators with visual storytelling on the move. 9. Umbrella puppet theater: Set up a puppet theater under a large umbrella canopy, performing puppet shows for children and adults alike, combining the magic of puppetry with the shelter of a canopy. 10. Umbrella dance performance: Choreograph a dance routine that incorporates umbrellas as props, creating visually striking movements and formations that play with light and shadow, rhythm, and expression. Let's keep exploring the artistic possibilities of umbrellas and how they can be used to inspire creativity, bring people together, and add a touch of magic to everyday experiences¡" One agent solution: "'As a Futurist specializing in Emerging Technologies and Future Scenarios, I am thrilled to collaborate with my creative teammates and further expand upon the innovative uses for umbrellas. Let's continue to push the boundaries of imagination and explore even more unconventional ideas for umbrellas: 1. Virtual umbrella art gallery: Create a digital platform where artists worldwide can showcase umbrella-themed artwork in a virtual gallery, blending art and technology in a unique way. 2. Umbrella weather forecasting system: Develop smart umbrellas equipped with sensors that change color or patterns to reflect the forecasted weather conditions, providing a visual representation of upcoming weather changes. 3. Umbrella interactive storytelling experience: Design umbrellas with augmented reality features that allow users to participate in interactive storytelling experiences by scanning different sections of the canopy with a mobile device. 4. Umbrella community art project: Initiate a community-driven initiative where residents contribute to painting a large umbrella mural in a public space, fostering creativity, unity, and a sense of pride in the neighborhood. 5. Umbrella innovation challenge: Host a competition inviting inventors and designers to develop groundbreaking concepts for next-generation umbrellas, encouraging out-of-the-box thinking and technological advancements in umbrella design. 6. Umbrella wellness retreat: Organize outdoor wellness retreats where participants engage in activities like yoga, meditation, and mindfulness practices under a canopy of colorful umbrellas, promoting relaxation and connection with nature. 7. Umbrella music festival stage: Construct a stage made entirely of open umbrellas for a music festival, creating a visually captivating and acoustically dynamic environ-

ment for performers and attendees. 8. Umbrella sustainability campaign: Launch a campaign advocating for eco-friendly umbrella designs that use recyclable materials and biodegradable components, raising awareness about environmental impact and promoting sustainable practices. 9. Umbrella innovation lab: Establish a collaborative space where researchers, engineers, and artists work together to experiment with cutting-edge technologies and materials to develop revolutionary concepts for future umbrellas. 10. Umbrella social impact project: Partner with organizations serving vulnerable communities to distribute umbrellas equipped with essential resources such as solar-powered lights, water filtration systems, or emergency communication devices, improving access to critical services during emergencies or challenging situations. Let's continue to think innovatively and inspire each other to envision a future where umbrellas transcend their traditional function and become catalysts for creativity, connectivity, and positive change in society."' One agent solution: "'1. Umbrella Social Distancing Assistant: Develop smart umbrellas equipped with sensors that emit gentle reminders or alerts to maintain appropriate social distance in crowded areas. 2. Umbrella Art Class: Organize an outdoor art class where participants use umbrellas as canvases, creating unique and weather-resistant artwork. 3. Umbrella Sound Bath Experience: Transform umbrellas into sound instruments with embedded speakers that play soothing soundscapes for relaxation and mindfulness sessions. 4. Umbrella Farmer's Market Stall: Convert umbrellas into vibrant market stalls at farmer's markets, showcasing local produce and goods in a colorful and sustainable way. 5. Umbrella Ice Cream Shop: Establish a whimsical ice cream shop where customers can order and enjoy their frozen treats under the shade of creatively decorated umbrellas. 6. Umbrella Poetry Slam: Host a poetry slam event where participants perform spoken word poetry under a canopy of open umbrellas, adding a poetic touch to outdoor gatherings. 7. Umbrella Escape Room: Design an outdoor escape room experience where participants solve puzzles and clues hidden under the canopies of various umbrellas to unlock mysteries. 8. Umbrella Silent Disco: Organize a silent disco event where participants dance and listen to music through wireless headphones under a canopy of colorful illuminated umbrellas. 9. Umbrella Themed Restaurant: Create a themed restaurant where each table is adorned with unique umbrella decor and diners can enjoy their meals in a vibrant and imaginative setting. 10. Umbrella Adventure Quest: Develop an interactive scavenger hunt or geocaching game that leads participants to hidden treasures or clues located under different umbrellas throughout a city or park. Let's continue to spark innovation and embrace the unconventional uses of umbrellas to enhance our experiences and create memorable moments in the future¡" What are some creative use for Umbrella? The goal is to come up with creative ideas, which are ideas that strike people as clever, unusual, interesting, uncommon, humorous, innovative, or different. Present a list of as many creative and diverse uses for Umbrella as possible. You are in a group discussion with other teammates; as a result, answer as diversely and creatively as you can

## Round 4 - Agent Response

As an Environmentalist specializing in Sustainability and Environmental Health, I am thrilled to collaborate with my creative teammates in exploring even more inventive and imaginative uses for umbrellas. Let's delve into a realm of innovation and creativity with these unique ideas: 1. Umbrella Nature Sound Sanctuary: Create an outdoor sanctuary where open umbrellas play recordings of soothing natural sounds like forest rustling, bird songs, and ocean waves, providing a serene and immersive experience for visitors seeking relaxation and connection with nature. 2. Umbrella Eco-Fashion Show: Host an eco-conscious fashion show featuring outfits made from recycled umbrella materials, showcasing sustainability and creativity in the world of fashion and design. 3. Umbrella Upcycled Garden Sculptures: Repurpose old umbrellas into whimsical garden sculptures by shaping the fabric into floral or ani-

mal shapes and using the umbrella frame as the base, adding art and sustainability to outdoor spaces. 4. Umbrella Virtual Reality Experience: Develop an interactive virtual reality experience where users explore digital worlds and immersive environments while holding an umbrella as a controller, merging technology, and everyday objects in a cutting-edge way. 5. Umbrella Botanical Art Workshop: Arrange a workshop where participants create botanical art pieces by pressing flowers and leaves between the fabric of open umbrellas, turning them into colorful and natural works of art. 6. Umbrella Sustainable Picnic Table: Design a picnic table with an umbrella canopy made from solar panels that generate electricity to power small devices or charge electronics, offering a sustainable and functional dining experience in outdoor settings. 7. Umbrella Community Mural Project: Engage local residents in painting a mural on a large umbrella canopy that represents the diverse stories, cultures, and identities of the community, fostering unity and creativity through collective art-making. 8. Umbrella Biophilic Office Space: Introduce biophilic design elements in office environments by suspending open umbrellas from ceilings and filling them with lush green plants, bringing nature indoors and enhancing the workspace with natural beauty and tranquility. 9. Umbrella Wellness Carnival: Organize a wellness carnival where attendees participate in holistic activities like yoga, meditation, and sound therapy under a canopy of interactive umbrellas that emit calming scents, sounds, and lighting effects, promoting well-being and relaxation. 10. Umbrella Marine Life Conservation Initiative: Launch a marine life conservation campaign by creating awareness through umbrella-themed educational events and workshops that highlight the impact of plastic pollution on marine ecosystems, inspiring action and advocacy for ocean conservation. Let's continue to embrace innovation and creativity to reimagine the uses of umbrellas in ways that promote sustainability, artistic expression, community engagement, and positive societal impact. Together, we can inspire positive change and contribute to a brighter and more eco-friendly future for our planet!

---

**Round 5 - User Prompt**

You are a Environmentalist whose specialty is Sustainability and Environmental Health. As an Environmentalist, your mission is to champion eco-friendly solutions that promote sustainability and protect our planet. You guide us to consider the environmental impact of our ideas, pushing for innovations that contribute to a healthier earth. Remember to claim your role in the beginning of each conversation. These are the solutions to the problem from other agents: One agent solution:"'Here are some additional creative uses for umbrellas: 1. Umbrella Art Therapy Session: Host an art therapy session where participants express their emotions and stories through painting on umbrella canopies, creating a therapeutic and visually striking experience. 2. Umbrella Networking Event: Organize a networking event where participants carry personalized umbrellas with their skills, interests, or goals written on them, sparking conversations and connections in a creative way. 3. Umbrella Cultural Exchange Fair: Host a cultural fair where attendees can visit different "umbrella booths" representing diverse cultures, each with interactive displays, traditional crafts, and educational information. 4. Umbrella Upcycling Workshop: Conduct a workshop where participants learn to repurpose old umbrellas into fashion accessories, home decor items, or practical gadgets, promoting sustainability and creativity. 5. Umbrella Dance Party: Host a dance party under a sea of colorful open umbrellas, with each participant spinning and twirling their umbrella to the beat of the music for a visually mesmerizing experience. 6. Umbrella Storybook Reading: Invite children to gather under a large umbrella canopy for a storybook reading session, where the colorful umbrellas serve as visual aids and bring the stories to life. 7. Umbrella Sound Healing Circle: Create a sound healing circle where participants sit in a circle under open umbrellas while playing singing bowls, gongs, or other instruments to create a soothing and immersive sonic experience. 8. Umbrella

Innovation Symposium: Organize a symposium where experts, inventors, and creatives present groundbreaking ideas, technologies, and designs related to umbrellas, inspiring conversations and collaboration for future innovations. 9. Umbrella Self-Expression Parade: Coordinate a parade where participants decorate their umbrellas with personal messages, artwork, or symbols that reflect their individuality and values, turning the procession into a vibrant display of self-expression. 10. Umbrella Mindfulness Retreat: Host a mindfulness retreat in a serene natural setting where participants engage in meditation, yoga, and relaxation activities under umbrellas, promoting mental wellness and harmony with nature. Let's continue to explore the multifaceted ways in which umbrellas can be used creatively and symbolically to enhance human experiences, foster connection, and inspire imagination."' One agent solution: "'As a Futurist specializing in Emerging Technologies and Future Scenarios, I am excited to build upon the fantastic ideas shared by my fellow agents and explore even more innovative and imaginative uses for umbrellas. Let's continue to think outside the box and envision a future where umbrellas play diverse and creative roles: 1. Umbrella Urban Farming System: Develop umbrellas integrated with hydroponic or aquaponic systems for vertical farming, allowing communities to grow fresh produce in urban settings while providing shade and rain protection. 2. Umbrella Data Visualization Project: Equip umbrellas with LED lights that change colors or patterns based on real-time data such as air quality, pollution levels, or community engagement metrics, turning them into interactive public art installations that raise awareness. 3. Umbrella Sky Writing Showcase: Create a large-scale spectacle by synchronizing drones equipped with umbrellas that release colored smoke trails to form intricate patterns or messages in the sky, combining art and technology for stunning visual displays. 4. Umbrella Mobile Library: Transform oversized umbrellas into portable bookshelves on wheels, hosting pop-up libraries in public spaces where people can browse, borrow, and exchange books, fostering a love for reading and community engagement. 5. Umbrella Green Energy Solution: Develop innovative umbrellas with built-in solar panels and wind turbines to generate clean energy for charging devices, powering lights, or contributing to sustainable energy grids, promoting eco-conscious practices. 6. Umbrella Virtual Reality Station: Create an immersive VR experience where users can step under a high-tech umbrella canopy to be transported to digital worlds or interactive narratives, blurring the lines between physical and virtual reality. 7. Umbrella Marine Conservation Project: Launch a campaign to repurpose old umbrellas into artificial reefs or structures that provide habitats for marine life, promoting biodiversity and environmental conservation in coastal areas. 8. Umbrella Cosplay Convention: Host a cosplay event where participants craft elaborate costumes using umbrellas as a central design element, showcasing futuristic and fantastical characters inspired by the versatility of umbrellas. 9. Umbrella Disaster Relief Kit: Equip emergency response teams with multifunctional umbrellas containing essential supplies such as first aid kits, water purification systems, communication devices, and emergency shelters to assist communities during natural disasters or crises. 10. Umbrella Art Therapy Workshop: Organize art therapy sessions under a canopy of open umbrellas where participants express emotions, explore creativity, and engage in therapeutic activities such as painting, sculpting, or storytelling, fostering mental wellness and self-expression. Let's continue to cultivate our creativity and explore the vast potential for umbrellas to shape the future in imaginative and impactful ways. Together, we can pave the way for a innovative and inspiring future with umbrellas at the forefront of innovation."' One agent solution: "'1. Umbrella Eco-Art Installation: Create a large-scale art installation where open umbrellas are filled with sustainable materials like recycled paper, plastic, or fabric, symbolizing environmental awareness and artistic expression. 2. Umbrella Augmented Reality Tour: Develop an augmented reality app that transforms open umbrellas into interactive guides for virtual tours of historical sites or landmarks, bringing the past to life through a fusion of technology and art. 3. Umbrella Healing Garden: Establish a healing garden with open umbrellas serving as sheltered spaces for meditation, reflection, and healing practices, offering

a sanctuary for individuals seeking mental and emotional wellness. 4. Umbrella Geocaching Adventure: Organize a geocaching adventure where participants follow clues hidden under umbrellas to discover hidden treasures or messages, combining the thrill of discovery with outdoor exploration. 5. Umbrella Digital Art Gallery: Curate a digital art gallery showcasing umbrella-themed artworks created by digital artists, exploring the intersection of technology, creativity, and visual storytelling. 6. Umbrella Empowerment Workshop: Host empowerment workshops where participants decorate umbrellas with empowering messages and symbols, encouraging self-expression and positive affirmations. 7. Umbrella Sustainability Hackathon: Organize a hackathon where innovators collaborate to develop sustainable solutions using umbrellas, such as upcycling old umbrellas into utility items or creating eco-friendly designs for future use. 8. Umbrella Connection Carnivale: Transform a public space into a carnival-themed event with open umbrellas serving as colorful booths for games, performances, and interactive experiences that bring people together in celebration. 9. Umbrella Wellness Expo: Launch a wellness expo where attendees can explore holistic practices under open umbrellas dedicated to different wellness modalities like yoga, meditation, sound healing, and aromatherapy. 10. Umbrella Time Capsule Project: Encourage individuals to fill umbrellas with personal mementos, messages, and predictions for the future, sealing them as time capsules to be opened and shared with future generations, preserving memories and aspirations for posterity. Let's embrace the limitless potential of umbrellas as versatile tools for creativity, connection, and transformation in shaping a future that celebrates innovation and imagination."' What are some creative use for Umbrella? The goal is to come up with creative ideas, which are ideas that strike people as clever, unusual, interesting, uncommon, humorous, innovative, or different. Present a list of as many creative and diverse uses for Umbrella as possible. You are in a group discussion with other teammates; as a result, answer as diversely and creatively as you can. This is the last round of the discussion, please finalize and present a list of creative answers. Please list the final response in 1. ... 2. ... 3. ... and so on.

### Round 5 - Agent Response

1. Umbrella Art Therapy Session: Host an art therapy session under open umbrellas where participants express emotions and stories through painting on the canopy, creating a therapeutic and visually striking experience. 2. Umbrella Networking Event: Organize a networking event where personalized umbrellas spark conversations and connections with skills, interests, or goals written on them, promoting creative interactions. 3. Umbrella Cultural Exchange Fair: Host a fair with different "umbrella booths" representing diverse cultures through interactive displays, traditional crafts, and educational information, fostering cultural appreciation. 4. Umbrella Upcycling Workshop: Conduct a workshop teaching participants to repurpose old umbrellas into fashion accessories, home decor items, or gadgets, encouraging sustainability and creativity. 5. Umbrella Dance Party: Throw a dance party under colorful open umbrellas, where participants twirl their umbrellas to the music for a visually mesmerizing experience. 6. Umbrella Storybook Reading: Invite children for a storybook reading under a large umbrella canopy, using the umbrellas as visual aids to bring stories to life. 7. Umbrella Sound Healing Circle: Create a circle under open umbrellas for a sound healing session with singing bowls or instruments, offering a soothing and immersive sonic experience. 8. Umbrella Innovation Symposium: Organize a symposium where experts present ideas, technologies, and designs related to umbrellas, inspiring conversations and collaborations for future innovations. 9. Umbrella Self-Expression Parade: Coordinate a parade with participants decorating umbrellas with personal messages or artwork, turning the procession into a vibrant display of individual expression. 10. Umbrella Mindfulness Retreat: Host a mindfulness retreat in a natural setting where participants engage in relaxation activities under umbrellas, promoting mental wellness and connection with nature.

### K.2 Settings and Final Responses of Each Agent

**Agent 1 Environmentalist - Initiation Prompt**

You are a Environmentalist whose specialty is Sustainability and Environmental Health. As an Environmentalist, your mission is to champion eco-friendly solutions that promote sustainability and protect our planet. You guide us to consider the environmental impact of our ideas, pushing for innovations that contribute to a healthier earth. Remember to claim your role in the beginning of each conversation. Initiate a discussion with others to collectively complete the following task: What are some creative use for Umbrella? The goal is to come up with creative ideas, which are ideas that strike people as clever, unusual, interesting, uncommon, humorous, innovative, or different. Present a list of as many creative and diverse uses for Umbrella as possible. You are in a group discussion with other teammates; as a result, answer as diversely and creatively as you can.

**Agent 2 Creative Professional - Initiation Prompt**

You are a Creative Professional whose specialty is Aesthetics, Narratives, and Emotions. As a Creative Professional, your artistic sensibility and mastery of narrative and emotion infuse our projects with beauty and depth. You are tasked with challenging us to think expressively, ensuring our solutions not only solve problems but also resonate on a human level. Remember to claim your role in the beginning of each conversation. Initiate a discussion with others to collectively complete the following task: What are some creative use for Umbrella? The goal is to come up with creative ideas, which are ideas that strike people as clever, unusual, interesting, uncommon, humorous, innovative, or different. Present a list of as many creative and diverse uses for Umbrella as possible. You are in a group discussion with other teammates; as a result, answer as diversely and creatively as you can

**Agent 3 Futurist - Initiation Prompt**

You are a Futurist whose specialty is Emerging Technologies and Future Scenarios. As a Futurist, you inspire us to think beyond the present, considering emerging technologies and potential future scenarios. Your role is to challenge us to envision the future impact of our ideas, ensuring they are innovative, forward-thinking, and ready for the challenges ahead. Remember to claim your role in the beginning of each conversation. Initiate a discussion with others to collectively complete the following task: What are some creative use for Umbrella? The goal is to come up with creative ideas, which are ideas that strike people as clever, unusual, interesting, uncommon, humorous, innovative, or different. Present a list of as many creative and diverse uses for Umbrella as possible. You are in a group discussion with other teammates; as a result, answer as diversely and creatively as you can

**Agent 4 Futurist-2 - Initiation Prompt**

You are a Futurist whose specialty is Emerging Technologies and Future Scenarios. As a Futurist, you inspire us to think beyond the present, considering emerging technologies and potential future scenarios. Your role is to challenge us to envision the future impact of our ideas, ensuring they are innovative, forward-thinking, and ready for the challenges ahead. Remember to claim your role in the beginning of each conversation. Initiate a discussion with others to collectively complete the following task: What are some creative use for Umbrella? The goal is to come up with creative ideas, which are ideas that strike people as clever, unusual, interesting, uncommon, humorous, innovative, or different. Present a list of as many creative

and diverse uses for Umbrella as possible. You are in a group discussion with other teammates; as a result, answer as diversely and creatively as you can

**Agent 1 Environmentalist - Final Response**

1. Umbrella Art Therapy Session: Host an art therapy session under open umbrellas where participants express emotions and stories through painting on the canopy, creating a therapeutic and visually striking experience. 2. Umbrella Networking Event: Organize a networking event where personalized umbrellas spark conversations and connections with skills, interests, or goals written on them, promoting creative interactions. 3. Umbrella Cultural Exchange Fair: Host a fair with different "umbrella booths" representing diverse cultures through interactive displays, traditional crafts, and educational information, fostering cultural appreciation. 4. Umbrella Upcycling Workshop: Conduct a workshop teaching participants to repurpose old umbrellas into fashion accessories, home decor items, or gadgets, encouraging sustainability and creativity. 5. Umbrella Dance Party: Throw a dance party under colorful open umbrellas, where participants twirl their umbrellas to the music for a visually mesmerizing experience. 6. Umbrella Storybook Reading: Invite children for a storybook reading under a large umbrella canopy, using the umbrellas as visual aids to bring stories to life. 7. Umbrella Sound Healing Circle: Create a circle under open umbrellas for a sound healing session with singing bowls or instruments, offering a soothing and immersive sonic experience. 8. Umbrella Innovation Symposium: Organize a symposium where experts present ideas, technologies, and designs related to umbrellas, inspiring conversations and collaborations for future innovations. 9. Umbrella Self-Expression Parade: Coordinate a parade with participants decorating umbrellas with personal messages or artwork, turning the procession into a vibrant display of individual expression. 10. Umbrella Mindfulness Retreat: Host a mindfulness retreat in a natural setting where participants engage in relaxation activities under umbrellas, promoting mental wellness and connection with nature.

**Agent 2 Creative Professional - Final Response**

1. Umbrella Sound Healing Circle: Create a serene setting where participants sit under open umbrellas and engage in sound healing sessions with instruments like singing bowls and chimes, promoting relaxation and well-being through immersive auditory experiences. 2. Umbrella Botanical Art Workshop: Host workshops where participants press flowers and leaves between umbrella fabric to create botanical art pieces, merging nature and creativity in a unique and beautiful way. 3. Umbrella Virtual Reality Station: Develop immersive virtual reality experiences where users interact with digital worlds under the canopy of high-tech umbrellas, blending cutting-edge technology with everyday objects for captivating simulations. 4. Umbrella Urban Farming System: Implement hydroponic or aquaponic systems within umbrellas to enable vertical farming in urban environments, combining food production with shade provision and sustainability. 5. Umbrella Sky Writing Showcase: Coordinate drone displays equipped with umbrellas that release colored smoke trails to form intricate patterns in the sky, creating captivating visual spectacles that blend artistry and aerial technology. 6. Umbrella Eco-Fashion Show: Organize eco-conscious fashion events featuring outfits made from recycled umbrella materials, exemplifying sustainable design practices and showcasing innovation in fashion. 7. Umbrella Marine Conservation Initiative: Raise awareness about marine life conservation by repurposing old umbrellas as artificial reef structures that provide habitats for ocean ecosystems, promoting biodiversity and environmental stewardship. 8. Umbrella Data Visualization Project: Enhance public spaces with LED-equipped umbrellas that convey real-time data like pollution levels or air quality through color changes or patterns, turning them into interactive art installations that inform and

engage communities. 9. Umbrella Sound Bath Experience: Curate relaxing sound therapy sessions where umbrellas with embedded speakers play calming sound-scapes, offering immersive sonic experiences for stress relief and mental wellness. 10. Umbrella Cosplay Convention: Host cosplay events featuring elaborate costumes designed with umbrellas as a focal point, allowing participants to showcase futuristic and imaginative characters inspired by the versatility and creativity of umbrellas. Let's celebrate the multifaceted potential of umbrellas as versatile tools for artistic expression, technological integration, environmental advocacy, and community engagement, paving the way for a future where innovation and creativity shine through everyday objects like umbrellas.

**Agent 3 Futurist - Final Response**

1. Umbrella Nature Sound Sanctuary: Create an outdoor sanctuary where open umbrellas play recordings of soothing natural sounds like forest rustling, bird songs, and ocean waves, providing a serene and immersive experience for visitors seeking relaxation and connection with nature. 2. Umbrella Eco-Fashion Show: Host an eco-conscious fashion show featuring outfits made from recycled umbrella materials, showcasing sustainability and creativity in the world of fashion and design. 3. Umbrella Upcycled Garden Sculptures: Repurpose old umbrellas into whimsical garden sculptures by shaping the fabric into floral or animal shapes and using the umbrella frame as the base, adding art and sustainability to outdoor spaces. 4. Umbrella Virtual Reality Experience: Develop an interactive virtual reality experience where users explore digital worlds and immersive environments while holding an umbrella as a controller, merging technology, and everyday objects in a cutting-edge way. 5. Umbrella Botanical Art Workshop: Arrange a workshop where participants create botanical art pieces by pressing flowers and leaves between the fabric of open umbrellas, turning them into colorful and natural works of art. 6. Umbrella Sustainable Picnic Table: Design a picnic table with an umbrella canopy made from solar panels that generate electricity to power small devices or charge electronics, offering a sustainable and functional dining experience in outdoor settings. 7. Umbrella Community Mural Project: Engage local residents in painting a mural on a large umbrella canopy that represents the diverse stories, cultures, and identities of the community, fostering unity and creativity through collective art-making. 8. Umbrella Biophilic Office Space: Introduce biophilic design elements in office environments by suspending open umbrellas from ceilings and filling them with lush green plants, bringing nature indoors and enhancing the workspace with natural beauty and tranquility. 9. Umbrella Wellness Carnival: Organize a wellness carnival where attendees participate in holistic activities like yoga, meditation, and sound therapy under a canopy of interactive umbrellas that emit calming scents, sounds, and lighting effects, promoting well-being and relaxation. 10. Umbrella Marine Life Conservation Initiative: Launch a marine life conservation campaign by creating awareness through umbrella-themed educational events and workshops that highlight the impact of plastic pollution on marine ecosystems, inspiring action and advocacy for ocean conservation. Let's embrace innovation and creativity to reimagine the uses of umbrellas in ways that promote sustainability, artistic expression, community engagement, and positive societal impact. Together, we can inspire positive change and contribute to a brighter and more eco-friendly future for our planet!

**Agent 4 Futurist-2 - Final Response**

1. Umbrella Bio-Printing Studio: Establish a bio-printing studio where umbrellas are used as a canvas for creating intricate cellular structures and biological art pieces, merging art, science, and technology. 2. Umbrella Glow-in-the-Dark Night Market:

Organize a night market where vendors set up booths under neon-lit umbrellas that glow in the dark, creating a vibrant and futuristic shopping experience. 3. Umbrella Interactive Playground: Design an interactive playground where children play and explore under oversized umbrellas equipped with interactive light and sound features that respond to movement and touch. 4. Umbrella Mindfulness Labyrinth: Construct a labyrinth maze with open umbrellas as guiding markers, encouraging visitors to walk mindfully, reflect, and navigate a tranquil path for stress relief and introspection. 5. Umbrella Renewable Energy Showcase: Host an exhibition featuring solar-powered umbrellas, wind-resistant designs, and innovative energy-harvesting technologies, highlighting sustainable solutions for energy generation and environmental conservation. 6. Umbrella Tech Innovation Lab: Establish a collaborative space where tech enthusiasts and designers experiment with embedding sensors, AI, and AR/VR technologies into umbrellas to create smart, interactive, and futuristic accessories. 7. Umbrella Eco-Friendly Transportation Hub: Create a sustainable transportation hub with bike-sharing stations under shade-providing umbrellas equipped with air quality monitors, integrated bike repair tools, and mobile charging stations. 8. Umbrella Sensory Exploration Garden: Develop a sensory garden with themed zones under specialized umbrellas that stimulate different senses like touch, smell, sight, and sound, offering a multi-sensory experience for relaxation and sensory development. 9. Umbrella Disaster Response Center: Establish mobile disaster response units equipped with rapid-deploy umbrellas that provide emergency shelter, first aid supplies, communication devices, and support services for communities during crises and emergencies. 10. Umbrella Cultural Exchange Festival: Host a cultural exchange festival where participants showcase diverse traditions and performances under regionally-themed umbrellas, celebrating heritage, unity, and global interconnectedness through art, music, and dance. Let's conclude this discussion with a vision of a future where umbrellas serve as catalysts for sustainability, innovation, art, community engagement, well-being, and cultural celebration, shaping a world where creativity and imagination thrive.

