# OpenReview forum: "LLM Discussion: Enhancing the Creativity of Large Language Models via Discussion Framework and Role-Play"
_colmweb.org/COLM/2024/Conference — COLM_

### Official Review · Reviewer_6CAt · 2024-05-07

**Rating:** 7
**Confidence:** 4
**Ethics Flag:** 1

**Summary:**

The authors present an LLM discussion framework for improving the creativity of LLM outputs by using multiple LLM agents with diversified role-playing instructions in a 3-step pipeline.

**Questions To Authors:**

- Why does your method often perform worse on fluency compared to some baselines? Is there a tradeoff for fluency vs diversity (similar to how you make this tradeoff when you change the temperature of generation)?
- Why is flexibility not a main metric? Isn't coming up with more different answers a pretty reasonable thing to evaluate? Or is the problem just that you don't have a good way to "deduplicate" different answers? Not sure if I'm misunderstanding what is meant by flexibility in your setup.
- Why are all the metrics 0 when you have 3 agents in fig 6?

**Reasons To Accept:**

The idea of giving different role-playing instructions to different agents seems intuitive and well-motivated, and the method appears empirically effective at increasing creativity. Additionally, although they are evaluating creativity which (I think) should be a fairly difficult thing to nail down evaluation-wise, the experimental setup and analysis seem reasonable and sound, which I think is a solid contribution in its own right (and could perhaps be highlighted more if the authors want to).

**Reasons To Reject:**

While the authors have shown that their method can improve the creativity of LLM outputs, it does appear to come at some cost to fluency. In LM generation one can typically increase diversity trivially at the cost of fluency by e.g. increasing the temperature. It's not immediately obvious to me that the proposed method is not just a fancy way of doing the same tradeoff, rather than "pushing the Pareto boundary" in some sense. While the current experimental results are fine, in my opinion this paper would be substantially stronger if the authors showed results on e.g., something like a creative problem solving task where the additional creativity actually results in something like higher success rate or accuracy, rather than solely improving creativity for its own sake.

---

> ### Author Rebuttal · Authors · 2024-05-30
>
> We sincerely thank the reviewer for the thorough and constructive comments. Please find the response to your questions below.
>
> > Temperature vs. creativity
>
> We additionally experimented with increased temperatures from 0.0 to 2.0 (1.0 was used for the main experiments) on AUT and present the results [here](https://shorturl.at/1ew8I). The results show that increasing temperature leads to higher diversity (a lower self-BLEU score) while not resulting in higher creativity scores. The model outputs nonsensical responses, e.g., unrecognizable symbols, with a temperature ≥ 1.6. This justifies the need to develop frameworks to improve LLM creativity. We will revise the paper to include this insight.
>
> > Fluency vs. diversity
>
> We would like to clarify that Fluency refers to the number of relevant responses instead of fluency in linguistics. Based on the results presented in the \[Temperature vs. creativity\] response, there is no clear tradeoff between Fluency and diversity -- increasing temperature leads to decreased self-BLEU score while Fluency is maintained.
>
> > Creative problem-solving
>
> This work aims to improve LLM creativity on widely recognized and validated benchmarks, while we agree that evaluating LLMs on creative problem-solving is a promising direction [1], which will be discussed in the revised paper.
>
> - [1] Tian et al. MacGyver: Are Large Language Models Creative Problem Solvers?
>
> > Why is Flexibility not a main metric?
>
> Flexibility measures the number of categories in the responses; for example, when asked to list uses for a fork, eating spaghetti and using it as a hanger would be two different categories while eating spaghetti and eating steak would be in the same category. We did not prioritize Flexibility as a main metric because (1) Unlike humans, LLMs can inherently generate a large number of responses, which inflates Flexibility, and (2) We prioritized Originality and Elaboration, which more directly reflect the model's creativity. The points align with [2-4].
>
> - [2] Guzik et al. The originality of machines: AI takes the Torrance Test. Journal of Creativity
> - [3] Haase et al. Artificial muses: Generative artificial intelligence chatbots have risen to human-level creativity. Journal of Creativity
> - [4] Hadas et al. Using Large Language Models to Evaluate Alternative Uses Task Flexibility Score. Thinking Skills and Creativity
>
> > Fig 6
>
> The scores in Fig 6 are normalized to [0, 1]. We will revise it to show the original scores to avoid confusion.

---

> > ### Comment · Reviewer_6CAt · 2024-06-04
> > **Thanks for your response**
> >
> > Thanks for the extra analysis on fluency vs diversity. I'll maintain my score of 7 and recommend acceptance.

---

### Official Review · Reviewer_6qWd · 2024-05-11

**Rating:** 6
**Confidence:** 4
**Ethics Flag:** 1

**Summary:**

The paper presents a method called LLM Discussion, which is a multi-turn multi-LLM method to attempt to extract more creative responses out of LLMs on common creativity "brainteasers" such as Alternative Uses. The work experiments on four types of tasks, and evaluates different conversational methods to extract creative answers. The LLM Discussion method is found to be more creative both through a robust human evaluation, and using an LLM as a judge, which is found to correlate satisfactorily with human judgments.

**Questions To Authors:**

Q1: Can you report the average length of response for each task and each approach? This would help explain to the reader whether length explains away the creativity difference or not. Why did you not include in your prompt an expected response length, to normalize for this important effect?

Q2: Besides the tasks you've collected, whose intended audience are humans under a time constraint, do you have a sense of what tasks could be more appropriate to evaluate LLMs' creativity abilities? Do you believe that the "number of turns" reflects a similar effect to human test-takers having "limited time" in TTCT?

Q3: Two of the core contributions in LLM Discussion seem to be: (1) the three phases, (2) the role-playing. Yet, neither of these two elements seems to have been ablated. Do you have experimental evidence showing that such elements indeed lead to improved creativity in the responses?

Q4: It is not 100% clear in the paper which LLM is used in experimentation. The draft clearly states which GPT model is used for auto-eval, but not for generation. Are you using GPT4? Are the different roles all played by the same underlying LLM, or different LLMs? Does that make a difference?

Q5: It is not 100% clear in the paper how the "Convergence Phase" is conducted. It seems logical that only one LLM would perform the final agglomeration based on the conversation so far. How do you pick which "role" gets to do the final response? (e.g., do you get the final response from the environmentalist or the social entrepreneur).

Q6: Temperature plays a strong role on the diverging nature of an answer. What temperature setting did you set the generation models with, and do you expect increased temperature to further improve creativity scores?

Minor notes:
- It seems a bit of a stretch to call tau correlations at ~0.3 "strong correlation"
- This reviewer appreciated the pun in the title of Section 6, kudos :)

**Reasons To Accept:**

- The work is thorough, considers prior both in education and recent CS work seriously, and discusses the implications and limitations in a nuanced and realistic way.
- The LLM Discussion framework -- although somewhat evident -- is well constructed, particularly with the concept of the three phases of discussion: initiation, discuss & convergence.
- The human annotation seems thorough, and the release of the collection of judgments might be useful to others studying creativity in similar settings.
- The paper is generally well-written, and although some details are lacking, these can easily be resolved in a round of revisions.

**Reasons To Reject:**

- Nothing serious, overall the work is creative and rigorous.
- The four tasks studied are known to be somewhat artificial. As the authors point out, the tests to evaluate creativity in humans might not be adequate to evaluate LLMs, since the latter can be infinitely verbose at very low cost. The authors point this out and attenuate the importance of certain TTCW metrics, yet the tasks themselves remain tasks that were created for humans, not LLMs.
- There is a strong correlation between response length and perceived creativity, both in human & LLM evals. The work did not directly control for length of response (which seems like it would have been very easy to do...) and the examples shown in Figure 4 unfortunately show that LLM Discussion outputs are much longer. Are the promising results therefore only reflecting the underlying signal that responses in that condition are significantly longer on average?

---

> ### Author Rebuttal · Authors · 2024-05-30
>
> We sincerely thank the reviewer for the thorough and constructive comments. Please find the response to your questions below.
> > Benchmarks
>
> While our evaluation uses existing tasks for human creativity, we agree that developing LLM-customized creativity benchmarks and metrics is a promising direction. We will discuss it in the revision.
>
> > \# turns vs. limited time
>
> We do not limit the output length in each turn, so it may not reflect limited time.
>
> > Response length
>
> Below, we report the average response length and their Pearson correlation coefficient $c$ to Originality/Elaboration: #words/$c_O$/$c_E$.
> ||AUT|Instances|Similarities|Scientific|
> |-|-|-|-|-|
> |Single Agent|14.59/0.22/0.47|1.48/0.25/0.36|13.26/0.20/0.57|23.07/0.02/0.33|
> |LLM Debate|23.48/0.24/0.57|1.95/0.21/0.42|21.55/0.10/0.46|37.41/0.04/0.19|
> |LLM Discussion|27.82/0.07/0.31|8.00/0.26/0.35|32.03/0.07/0.21|47.23/0.11/-0.05|
>
> The results show a weak correlation between response length and creativity scores. Also, we present [histograms](https://shorturl.at/sfrir) analyzing response lengths and scores, showing that our method achieves better scores across varying length intervals. Hence, our improved creativity should not simply be attributed to longer responses. We will revise the paper to include the analysis.
>
> > Ablation study
>
> As suggested, we evaluated LLM Debate + role-play (w/o discussion), and LLM Discussion w/o role-play. The results below show the efficacy of each component. We will include the ablation study in the revision.
> ||Originality|Elaboration|
> |-|-|-|
> |LLM Debate w role-play|3.62 |3.12|
> |LLM Discussion w/o role-play|3.50| 3.02|
> |LLM Discussion|3.83|3.10|
>
> > Which LLM
>
> We used gpt-3.5-turbo-0125 for all the methods, which will be clarified in the revision.
>
> > Which role performs convergence
>
> Each and every role produces its final responses. All the responses are graded separately and averaged to obtain the score. We will revise the paper to make this clear.
>
> > Temperature vs. creativity
>
> Please refer to the \[Temperature vs. creativity\] response to Reviewer 6CAt.
>
> > Kendall rank correlation coefficient
>
> We interpret Kendall's tau correlation based on [1-2], stating that a correlation greater than 0.3 is considered strong.
> - [1] Botsch. Chapter 12: Significance and measures of association. Scopes and methods of political science, 2011.
> - [2] Chiang and Lee. Can large language models be an alternative to human evaluations? Empirical Methods in Natural Language Processing, 2023.

---

### Official Review · Reviewer_nFPB · 2024-05-11

**Rating:** 7
**Confidence:** 4
**Ethics Flag:** 1

**Summary:**

The presented approach gets much better results on key measures of creativity compared to SOTA.

This paper is a well-written presentation of the novel LLM Discussion approach to generating creative responses on established creativity and scientific innovation benchmark tests.

The approach is a highly original simulation of "collaborative discussions with diversified peers" that involves (i) a three-phase, initiation-discussion-convergence framework for multi-turn dialogues designed to elicit diverse and creative responses from (ii) LLM-based agents, each of what are assigned distinct roles and are directed to generate answers building on the others' responses.

This framework, (i), is compared against single-LLM and multi-LLM approaches and outperforms these previous methods on several creativity metrics.

The role-playing LLM-based agents, (ii), cleverly leverages the ability of LLMs to imitate very different personalities in order to generate more diversity of ideas.

**Questions To Authors:**

* The space of possible prompts and  ways of combining the results of prompts is so broad that it seems unlikely these results are anywhere near the best that could be obtained with present models. Q1: Why, for instance, are the particular roles of "visionary millionaire, startup founder, social entrepreneur, creative professional, customer, environmentalist, digital nomad, industry insider, futurist" (which sounds like a list of people you might encounter at a party in Silicon Valley) chosen instead of, say, homemaker, mathematician, janitor, and professional actor? Or, say, all 16 Meyers-Briggs personality types? Or people from 20 diverse countries or time periods? There's no justification for this and no way of knowing what would be best a priori. At the moment there is so much low-hanging fruit that it is easy to come up with new approaches that get big gains, but it would be helpful to find some automated way of searching the space of possibilities more thoroughly.

* Q2: What might help explain why fluency and flexibility scores are lower for LLM Discussion than LLM Debate, even though these matter less than originality and elaboration as measures of creativity. Perhaps there would be some simple way to alter the prompts to boost these factors as well?

* Q3: how is the prompt in the convergence phase leading to "convergence" as opposed to simply enumerating answers from prior phase, given instruction to  "present a list"?  Did you experiment with an explicit prompt to create a summary or draw a conclusion?

**Reasons To Accept:**

* Ambitious objective to simulate "collaborative discussions with diversified peers" with novel design of 3-phase framework for discussions by role-playing LLM agents.
* Exciting results with strong gains over previous methods of generating creative answers.
* Clear delineation of where this approach contrasts with state of the art.
* Solid empirical evaluation methodology with established tests and metrics, engaging both automated and human judgements, as well as documented ablation tests, explaining development of specialized prompts, # discussion rounds and # LLM agents

**Reasons To Reject:**

* The authors do not discuss going beyond the role-playing agents as artificial people, and address other ways of coming up with better answers,.

* The paper does not address what is actually happening inside the model, that would also give deeper insights.

---

> ### Author Rebuttal · Authors · 2024-05-30
>
> We sincerely thank the reviewer for the thorough and constructive comments. Please find the response to your questions below.
>
> > The specific roles used
>
> We agree with the reviewer that systematically exploring diverse roles is a promising direction. This work makes the first attempt to integrate multi-LLM discussion and role-play; hence, we focused on proposing an automated pipeline using GPT-4 to generate diverse and detailed roles motivated by creative thinking (role-storming).
>
> > Qualitative analysis: what happens inside the model
>
> We appreciate the reviewer’s suggestion. We will revise the paper to include more qualitative results and analyses from three aspects:
>
> - Collaborative dynamics: Our framework stimulates a more collaborative tone, e.g., ”building on those ideas, …,” while LLM Debate does not follow up on each other’s ideas.
> - Role-specific responses: LLMs can stay in character, e.g., in an AUT scenario of an umbrella, the Futurist proposes integrating it with VR technology, while the Environmentalist suggests using it as a shelter for wildlife. In contrast, the baselines tend to suggest more general uses.
> - Specific details: LLM Discussion outperforms baselines in Elaboration by coming up with detailed and specific answers. When asked ”Name all the things you can think of that are used in culture” from Instances, ours answers Tattoos, Digital art, and Ethical fashion, while the baselines answer Art, Music, and Clothing.
>
> > Boost Fluency and Flexibility
>
> As suggested by the reviewer, we additionally experimented with asking the LLM Discussion agent to generate as many answers as possible (AMAP) in the convergence phase, dubbed Ours+AMAP. The results below show that the prompt can increase Fluency and Flexibility scores while maintaining Originality and Elaboration.
>
> |Benchmark|Method|Originality|Elaboration|Fluency|Flexibility|
> |-|-|-|-|-|-|
> |AUT|Ours|4.44|4.22|9.19|9.68|
> ||Ours+AMAP|4.40|4.23|9.38|9.58|
> |Instances|Ours|3.65|2.20|16.88|11.11|
> ||Ours+AMAP|3.65|2.22|22.06|11.08|
> |Similarities|Ours|3.29|2.52|7.27|8.14|
> ||Ours+AMAP|3.27|2.55|10.89|11.52|
> |Scientific|Ours|3.95|3.47|5.58|5.91|
> ||Ours+AMAP|3.89|3.32|7.63|8.08|
>
> > Convergence phase
>
> As the reviewer pointed out, the term "finalize" in our convergence prompt serves as an explicit prompt to conclude, while "present a list" facilitates parsing. The chat logs show that the number of answers is reduced after convergence since some similar answers are combined and some are filtered out.

---

> > ### Comment · Reviewer_nFPB · 2024-06-07
> >
> > No change to the score of 7 recommending acceptance.
> >
> > The new table with the AMAP results ought to be included in the paper.
> >
> > At a core level what is happening internally is that the diversity of responses from a single prompt is limited.  For the approach taken in this paper, the setup of pretending to be various types of people is one way to add variety into the prompts, yielding extra originality. The setup of taking on different roles throws the generation into somewhat different distributions of training data, yielding different responses. It may be that even more diversity of responses would emerge if there were  increased diversity of the types of people being simulated or otherwise broadening of the diversity of the prompts.  Perhaps translating into different languages and asking the questions would be one way of getting answers from a very different part of the training distribution.

---

### Official Review · Reviewer_NP7f · 2024-05-22

**Rating:** 6
**Confidence:** 5
**Ethics Flag:** 1

**Summary:**

This paper proposes an approach to enhancing the creativity of large language models (LLMs) by facilitating multi-agent discussions and incorporating role-playing elements. The key ideas of a three-phase discussion framework and role assignment are well-motivated and explained. The authors conduct evaluations across 4 creativity benchmarks, comparing their proposed method against baselines. The results demonstrate improvements in some metrics of TTCT like originality and elaboration, suggesting that the proposed approach can indeed induce more creative and novel responses from LLMs. Overall, this work presents a relatively convincing approach to an important problem in AI, with findings that are likely to stimulate further research in this area.

**Questions To Authors:**

Why AUT ? There is mounting evidence that GPT4 is good at this so to benchmark you should focus on complex creative tasks
https://arxiv.org/abs/2303.12003
https://news.uark.edu/articles/69688/ai-outperforms-humans-in-standardized-tests-of-creative-potential

**Reasons To Accept:**

1) Clever use of multi agent discussion and roleplaying in open domain NLG task
2) Good evaluation with LLM and humans and also showing correlation between each

**Reasons To Reject:**

While the paper presents an interesting approach and promising results, there are a few key concerns that should be addressed before it is ready for publication

1) The role-playing component is a novel contribution, but the paper lacks deeper analysis on the specific roles used, how they were chosen/generated, and whether these roles may introduce unintended biases or perspectives that could skew the results.The role-playing aims to induce diverse perspectives, but the paper does not discuss how the chosen roles represent or account for diversity across different demographics, cultures, domains, etc. A more critical examination of this aspect is needed.

2) The task chosen are relatively old and sentence level so not very clear if this approach would be generalizable for other creative tasks such as design, ideation, creative writing, generating metaphors etc

3) The results are primarily quantitative, but a deeper qualitative analysis examining the generated responses, the nature of conversations/discussions, and specific strengths/weaknesses could provide valuable insights.

4) Theoretical grounding: While the motivations are clear, the paper could benefit from stronger theoretical foundations explaining why multi-agent discussion and role-playing are expected to enhance LLM creativity from a cognitive/psychological perspective.

---

> ### Author Rebuttal · Authors · 2024-05-30
>
> We sincerely thank the reviewer for the thorough and constructive comments. Please find the response to your questions below.
>
> > Role selection & generation
>
> We made the first attempt to integrate multi-LLM discussion and role-play; hence, we propose an automated pipeline using GPT-4 to generate diverse and detailed role descriptions motivated by creative thinking (role-storming). We will include the dialog with GPT-4 to generate roles in the revision. We agree with the reviewer that systematically exploring diverse roles is a promising direction.
>
> > Qualitative analysis
>
> Please refer to the \[Qualitative analysis: what happens inside the model\] response to Reviewer nFPB.
>
> > Theoretical foundations of multi-agent discussion and role-playing
>
> We are motivated by [1-4], suggesting that discussing with individuals from diverse backgrounds and perspectives can enhance creativity. Also, [5] shows that role-playing helps participants think from various perspectives, leading to novel solutions. We will revise the paper to discuss these works.
>
> - [1] Han, et al. Is group work beneficial for producing creative designs in STEM design education? International Journal of Technology and Design Education, 2022.
> - [2] Paulus and Nijstad, eds. Group Creativity: Innovation Through Collaboration. Oxford University Press, 2003.
> - [3] McGrath. Groups: Interaction and Performance. Englewood Cliffs, NJ: Prentice-Hall, 1984.
> - [4] Sutton and Hargadon. Brainstorming groups in context: Effectiveness in a product design firm. Administrative Science Quarterly, 1996.
> - [5] Karwowski and Soszynski. How to develop creative imagination?: Assumptions, aims and effectiveness of Role Play Training in Creativity (RPTC). Thinking Skills and Creativity 3.2, 2008.
>
> > Tasks chosen & why AUT? GPT4 is good at it
> - Our experiments use GPT-3.5, which has limited performance on AUT, and our framework can improve it.
> - We additionally evaluated single-agent GPT-4 on AUT, which achieves 3.83 in Originality and 3.74 in Elaboration, outperforming single-agent GPT-3.5 (O: 3.47, E: 3.08) aligning with the papers mentioned by the reviewer, yet underperforming ours with GPT-3.5 (O: 4.44, E: 4.22).
> - While AUT is widely adopted in creativity evaluation, we agree that merely using AUT may not be sufficient; hence, our evaluations considered three additional benchmarks, Instances, Similarities, and Scientific, to cover distinct aspects of creativity.
>
> We will revise the paper to include the above discussion.

---

> > ### Comment · Reviewer_NP7f · 2024-06-07
> > **Thank You**
> >
> > I will keep my scores. I don't think I am fully convinced at 3/4 answers

---

### Decision · Program_Chairs · 2024-07-10

**Decision:**

Accept

**Comment:**

The paper presents a novel LLM Discussion approach to enhance the creativity of large language models through multi-agent discussions and role-playing. Evaluations across four creativity benchmarks demonstrate improvements in originality and elaboration compared to single-LLM and existing multi-LLM frameworks. While some concerns are raised regarding role selection, potential biases, and the need for deeper qualitative analysis, the overall work is considered creative, rigorous, and likely to stimulate further research. I therefore recommend accepting this paper. There is also a related approach here https://arxiv.org/pdf/2310.00280 that the authors may find interesting.